# The nuclease EXO1 promotes genomic instability by degrading nascent DNA in BRCA-proficient cells

Alexandra Nusawardhana, Claudia M. Nicolae [ID] & George-Lucian Moldovan [ID] ✉

DNA repair genes are generally considered tumor suppressors, as their inactivation is observed in tumors and is associated with carcinogenesis. Mutations in BRCA1 and BRCA2 genes are observed in breast, ovarian, and other cancers. This results in defective homologous recombination DNA repair, as well as in degradation of nascent DNA during replication stress, catalyzed by exonucleases including EXO1 and MRE11. However, most tumors are BRCA pathway-proficient. Here, we show that EXO1 is overexpressed in a significant proportion of tumors. EXO1 overexpression causes the degradation of nascent DNA at both single stranded DNA (ssDNA) gaps and reversed replication forks. Importantly, this degradation occurs efficiently in BRCA-proficient cells, through cooperation with MRE11. This results in increased double strand break formation and hypersensitivity to genotoxic agents. We thus identify increased EXO1 activity as a mechanism of genomic instability similar to BRCA pathway inactivation, but occurring more frequently in tumors compared to BRCA inactivation.

Genomic instability is an enabling characteristic of carcinogenesis, allowing cancer cells to acquire new phenotypes and evolve constantly[1]. To ensure continuous proliferation, cancer cells need to replicate their DNA at appropriate rates. This exposes them to increased replication stress, caused by the encounter of the DNA replication machinery with unrepaired DNA lesions or by reduced availability of nucleotides, resulting in slowing or arrest of replication forks[2–4]. Cancer cells harness cellular genome stability mechanisms to control replication stress and may even use them to drive genome instability[5,6].

Stalled replication forks can be restarted through several mechanisms, primarily among those being fork repriming and fork reversal. Fork repriming entails the assembly of a new replication complex downstream of the stalled fork, initiated by the repriming activity of the PRIMPOL primase-polymerase[7–18]. This leaves behind a single stranded DNA (ssDNA) gap to be filled at a later time. In contrast, fork reversal involves the annealing of the nascent strands of the sister chromatids, catalyzed by the ZRANB3, HLTF, and SMARCAL1 translocases, generating a 4-way junction which stabilizes the fork and allows restart using the nascent strand of the sister chromatid as template[6,8,19–22].

EXO1 is a 5′−3′ exonuclease which participates in multiple DNA repair processes involving exonucleolytic processing of DNA lesions[23]. EXO1 plays an important role in long-range DNA resection at double stranded DNA breaks (DSBs)[24]. This process generates long single stranded DNA (ssDNA) overhangs which are subsequently coated by RAD51 molecules loaded by the BRCA2 protein. EXO1 is thus an important player in BRCA-mediated homologous recombination (HR) repair of DSBs. EXO1 also participates in DNA mismatch repair (MMR), by degrading the nascent DNA strand containing the mismatch, to allow repair DNA synthesis from the parental strand[25].

However, beyond these roles of EXO1 in promoting genomic stability, recent research has indicated that EXO1 activity can also promote genomic instability in certain genetic backgrounds. On reversed forks, BRCA-mediated loading of RAD51 is essential to protect the DSB end on the reversed arm formed by the nascent strand of the sister chromatids. In BRCA-mutant cells, this structure is degraded by nucleolytic activities, primarily those of MRE11 and EXO1. Indeed, it was proposed that one mechanism through which BRCA-deficient cells are sensitive to genotoxic cancer therapeutic drugs is the degradation of nascent DNA at reversed forks[6,8,19–22]. We

Department of Molecular and Precision Medicine, The Pennsylvania State University College of Medicine, Hershey, PA, USA. ✉e-mail: glm29@psu.edu

recently showed that EXO1 engages the reversed arm to degrade the nascent DNA in the 5′–3′ direction, while MRE11 degrades the complementary strand in the 3′–5′ direction[26].

Similarly, the filling of nascent strand ssDNA gaps is essential to suppress their nucleolytic expansion. Gap filling can be achieved by multiple mechanisms, including BRCA-mediated recombination using the nascent strand of the sister chromatid as a template, or by employing DNA polymerases able to bypass DNA lesions, a process termed translesion DNA synthesis (TLS). The inability to fill ssDNA gaps has been associated with genomic instability and chemosensitivity. Recent studies have proposed that accumulation of ssDNA gaps is one of the components causing increased sensitivity to cisplatin and PARP inhibitors in BRCA-deficient cells, since they get converted into DSBs, which are cytotoxic in these cells[8,11,26–38]. Moreover, concomitant inactivation of both the BRCA and TLS pathways synergize to provide increased chemosensitivity[10,11], further highlighting the importance of ssDNA gap repair.

We recently showed that, in BRCA-deficient cells, cisplatin-induced ssDNA gaps are exonucleolytically expanded by MRE11 and EXO1 in opposing directions, in a process reminiscent of the degradation of reversed forks[39,40]. We also found that unfilled ssDNA gaps are processed into DSBs, and exonucleolytic gap expansion is a critical intermediate in this processing, since inactivation of EXO1 or of MRE11 suppressed ssDNA gap-induced DSB formation. Importantly, EXO1 depletion also suppressed the cisplatin sensitivity of BRCA2-deficient cells. This suggests that EXO1-mediated expansion of ssDNA gaps contributes to cisplatin toxicity in these cells.

The role of the BRCA pathway in protecting nascent DNA against nucleolytic degradation at ssDNA gaps and reversed replication forks is considered a critical component of its tumor suppression activity. While BRCA inactivation is found in a subset of cancers, most tumors are thought to be BRCA-proficient. Thus, other genetic changes to drive genome instability through these mechanisms may exist in cancer cells. Here, we show that EXO1 is overexpressed in a significant proportion of cancers, including breast, hepatocellular, skin, testicular, and cervical cancers and EXO1 expression is associated with increased genomic alterations. We moreover show that EXO1 overexpression causes nascent strand degradation, at both ssDNA gaps and reversed forks, through its exonuclease catalytic activity. EXO1-mediated nascent strand degradation occurs efficiently in BRCA-proficient cells, through its cooperation with MRE11. This results in increased DSB formation and hypersensitivity to genotoxic agents. We thus identify increased EXO1 activity as a mechanism of genomic instability, similar to BRCA inactivation but occurring in a larger proportion of tumors.

## Results

### EXO1 is overexpressed in tumors which is associated with increased genome instability

To assess the relevance of EXO1 in cancer, we queried the TCGA database[41] for EXO1 alterations in tumor samples using the cBioPortal repository[42]. In line with previous studies[43–47], we found that EXO1 is predominantly overexpressed, through both gene amplification and increased mRNA levels without gene amplification, in all tumor datasets analyzed (Supplementary Fig. 1a). In contrast, there were very few instances of mutations, deletions, or reduced mRNA levels. These findings indicate that, despite the known role of EXO1 in DNA repair, increased EXO1 expression, rather than loss of EXO1 function, is found in cancer samples.

In particular, we found that alterations in EXO1 are most predominant in breast cancer, with ~22% of all breast invasive carcinoma samples in the TCGA PanCancer Atlas dataset showing such alterations. Of those, the vast majority represents EXO1 overexpression (~21%) compared to EXO1 mutation/deletion (~1%). To validate these findings, we investigated EXO1 expression in breast cancer cell lines.

Consistent with previous studies[45], we found lower EXO1 expression in the MCF10A normal epithelial mammary gland cell line, compared to BRCA pathway-proficient breast cancer cell lines T47D, MDA-MB-231, MCF7 and MDA-MB-468 (with significant variation of EXO1 expression within the breast cancer cell lines) (Supplementary Fig. 1b).

When we compared breast cancer subtypes, analyses of TCGA PanCancer Atlas breast invasive carcinoma samples revealed an over-representation of basal-like breast cancer in EXO1-overexpressing samples, to about ~50% of all cases in the dataset, compared to control (non EXO1-overexpressing) samples, of which only about ~8% were basal-like (Supplementary Fig. 2a). Basal-like breast cancer is an aggressive subtype of breast cancer, characterized by lack of expression of estrogen, progesterone or HER-2/neu hormones receptors (triple negative). It is considered that basal-like breast cancer tumors have a different etiology than hormone receptor-positives breast tumors, with genomic instability rather than hormone-induced proliferation being the major determinant[48–51]. Indeed, basal-like breast cancer is sometimes associated with BRCA1 inactivation. However, in the TCGA PanCancer Atlas breast invasive carcinoma dataset analyzed, only 3% of samples had BRCA1 mutations. We thus sought to investigate if, similar to BRCA-inactivated tumors, EXO1-overexpressing tumors show increased genomic instability. We found that EXO1 expression in breast tumors in this dataset is not associated with increased point mutation rates (Supplementary Fig. 2b). However, we found that EXO1 expression positively correlates with genome alterations (amplifications, deletions, translocations) (Supplementary Fig. 2c), suggesting that EXO1 promotes chromosomal structural instability and formation of chromosome structural variants, which are known to result from incorrect repair of DSBs.

Besides breast invasive carcinoma, the TCGA PanCancer Atlas analyses showed that EXO1 alterations were also prevalent in hepatobiliary cancer (with ~20% of cases showing EXO1 alterations), melanoma and testicular cancers (~16% each), and cervical cancer (~14%) (Supplementary Fig. 1a). Increased EXO1 expression was vastly predominant in these cancers as well, at the expense of mutations or deletions/downregulation. To further evaluate the connection between EXO1 overexpression and genomic instability, we expanded the mutational analysis to these cancers. We combined the breast, hepatobiliary, melanoma, testicular, and cervical tumor TCGA PanCancer Atlas datasets and evaluated the pooled dataset for genomic instability. We were able to confirm a correlation between EXO1 expression and genome alterations (Supplementary Fig. 2d).

As controls, we investigated how expression of other DNA repair genes affected genome alterations. We previously showed that PARP10 is overexpressed in breast and ovarian cancers[52]. We also showed that PARP10 interacts with the ubiquitin ligase RAD18 to promote the ubiquitination of PCNA, an event which is critical for TLS-mediated lesion bypass[53]. TLS is catalyzed by mutagenic polymerases and causes point mutagenesis[54,55]. In contrast to EXO1, the expression of PARP10, RAD18, or of the TLS polymerase POLH were not correlated with increased genome alteration (Supplementary Fig. 2e–g). Overall, these findings argue that increased EXO1 levels may be associated with increased genomic instability in cancer samples.

### EXO1 overexpression results in replication-associated DNA lesions upon replication stress

To directly test the impact of EXO1 expression levels on genomic instability, we stably overexpressed wildtype EXO1 (EXO1^OE) in BRCA pathway-proficient HeLa (cervical cancer) and U2OS (osteosarcoma) cells using a lentiviral expression system (Fig. 1a, b). As controls, we also obtained stable cell lines expressing the empty vector (EV) or overexpressing the EXO1 catalytic mutant D173A (EXO1-D173A^OE), which inactivates its exonuclease activity[56,57]. Both variants were expressed at similar levels to each other. Moreover, the EXO1

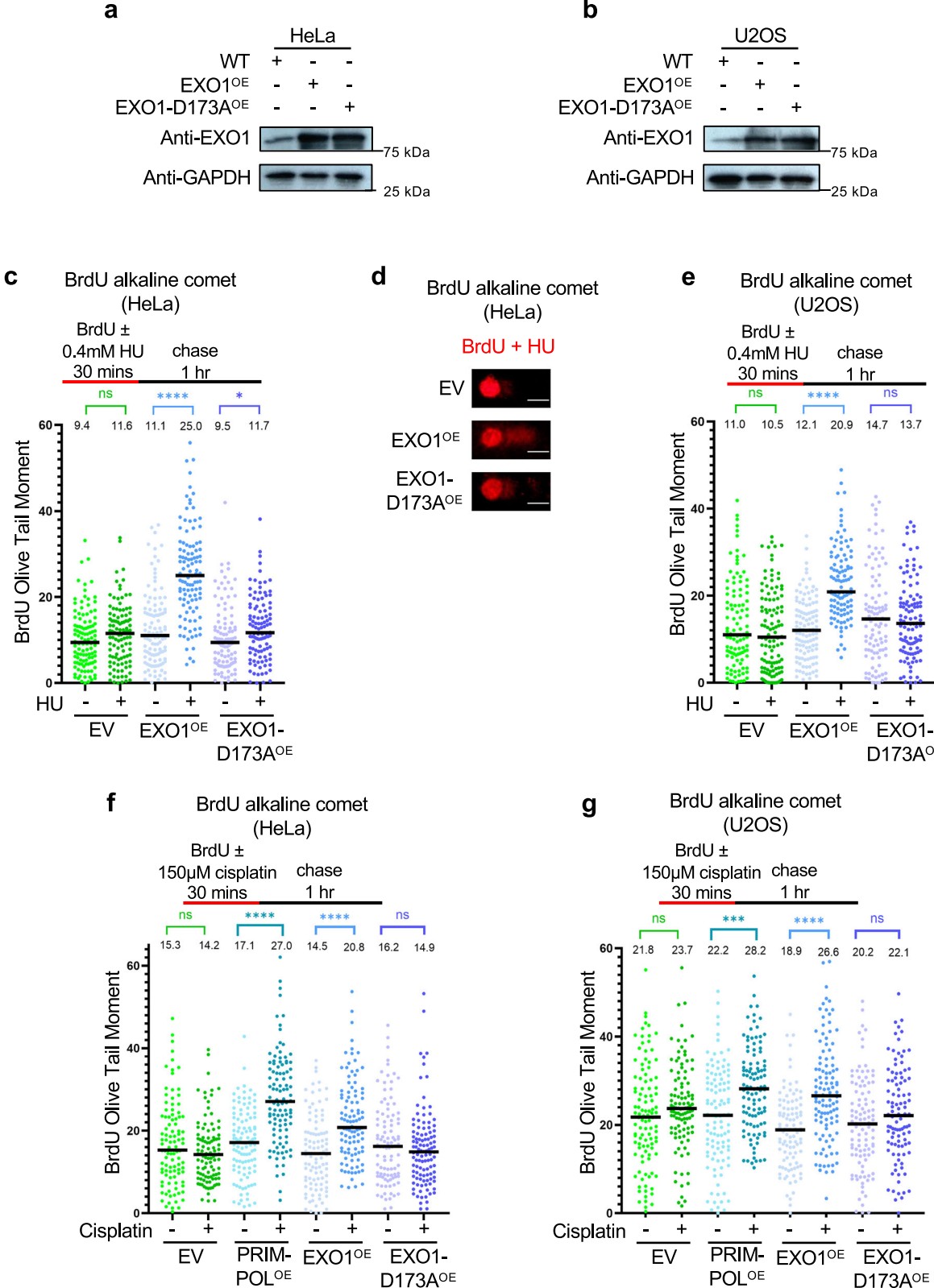

**Fig. 1 | EXO1 overexpression results in replication-associated DNA lesions upon replication stress.** Western blots showing the expression of EXO1 in control and EXO1-overexpressing HeLa (**a**) and U2OS (**b**) cells. **c–e** BrdU alkaline comet assay showing that overexpression of EXO1, but not of the D173A catalytic inactive mutant, causes accumulation of replication-associated single stranded DNA lesions upon treatment with 0.4 mM HU (**c–e**) or 150 μM cisplatin (**f, g**) in HeLa (**c, d, f**) and

U2OS (**e, g**). Representative micrographs with scale bars representing 10 μm (**d**) and quantifications (**c, e, f, g**) are shown. At least 100 nuclei were quantified for each condition. The median values are marked on the graph and listed at the top. Asterisks indicate statistical significance (Mann–Whitney, two-tailed). Schematic representations of the assay conditions are shown at the top. Source data are provided as a Source Data file.

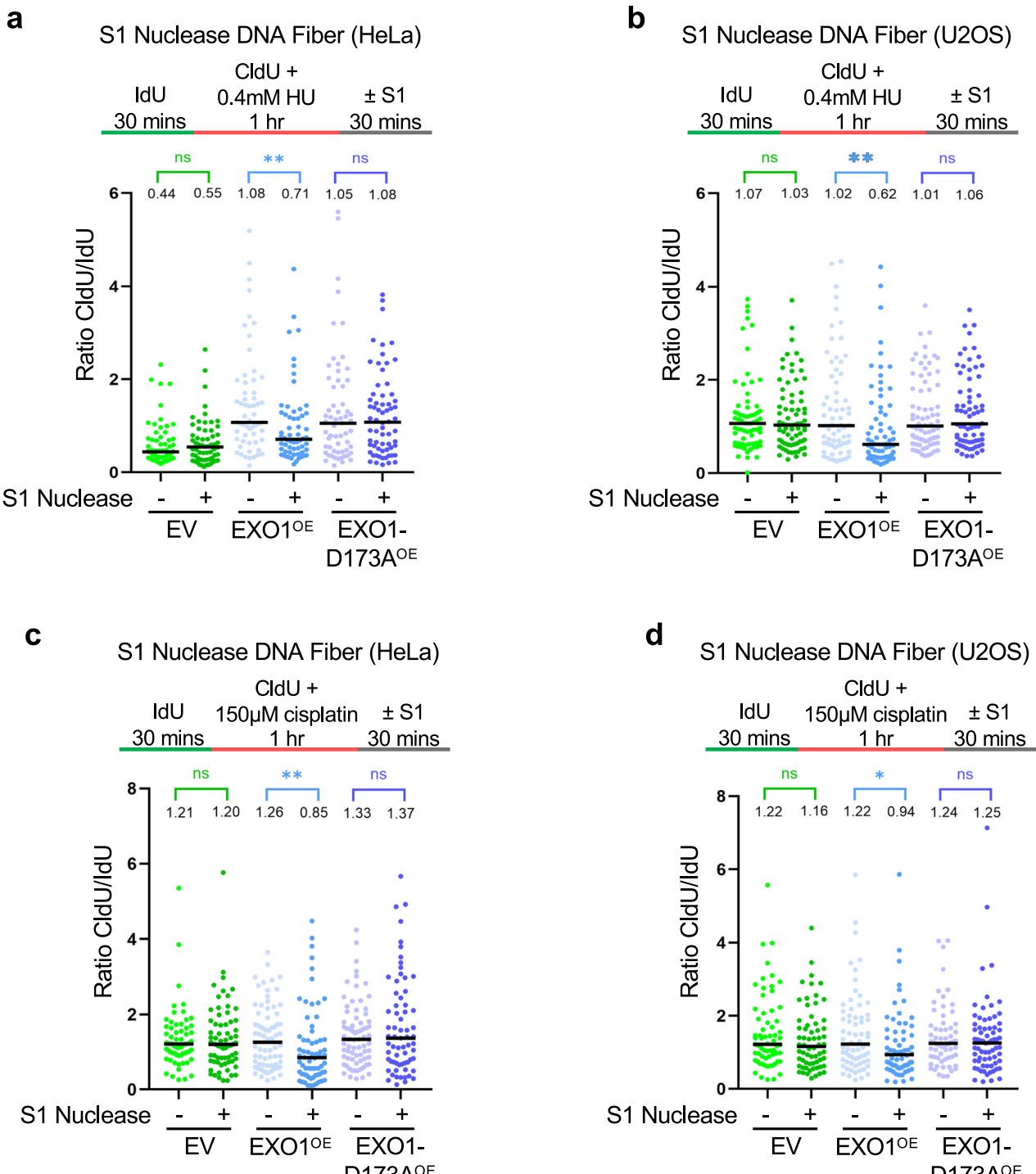

**Fig. 2 | EXO1 overexpression results in accumulation of ssDNA gaps in BRCA-proficient cells.** S1 nuclease DNA fiber combing assays showing that overexpression of EXO1, but not of the D173A catalytic inactive mutant, causes accumulation of nascent strand ssDNA gaps induced by treatment with 0.4 mM HU (**a**, **b**) or 150 μM cisplatin (**c**, **d**) in HeLa (**a**, **c**) and U2OS (**b**, **d**) cells. The ratio of CldU to IdU tract lengths is presented, with the median values marked on the graphs and listed at the top. At least 60 tracts were quantified for each sample. Asterisks indicate statistical significance (Mann–Whitney, two-tailed). Schematic representations of the assay conditions are shown at the top. Source data are provided as a Source Data file.

overexpression achieved in this experimental system is comparable to EXO1 expression in breast cancer cell lines (Supplementary Fig. 1b).

We previously showed that, upon replication stress, EXO1 degrades nascent DNA in BRCA-deficient cells, both at reversed replication forks[26], as well as upon fork restart by PRIMPOL and gap formation[39]. We thus first investigated if EXO1 overexpression causes replication-associated DNA lesions through its catalytic activity. We

employed the alkaline comet assay, which can detect single strand DNA lesions, as well as double stranded DNA breaks. To only detect replication-associated lesions, we treated cells with the thymidine analog BrdU to label replicating cells, followed by 1 hr growth in BrdU-free media ("chase") to allow the processing of normal replication intermediates such as DNA nicks on the lagging strand. During the BrdU labeling, we induced replication stress by treatment with a mild

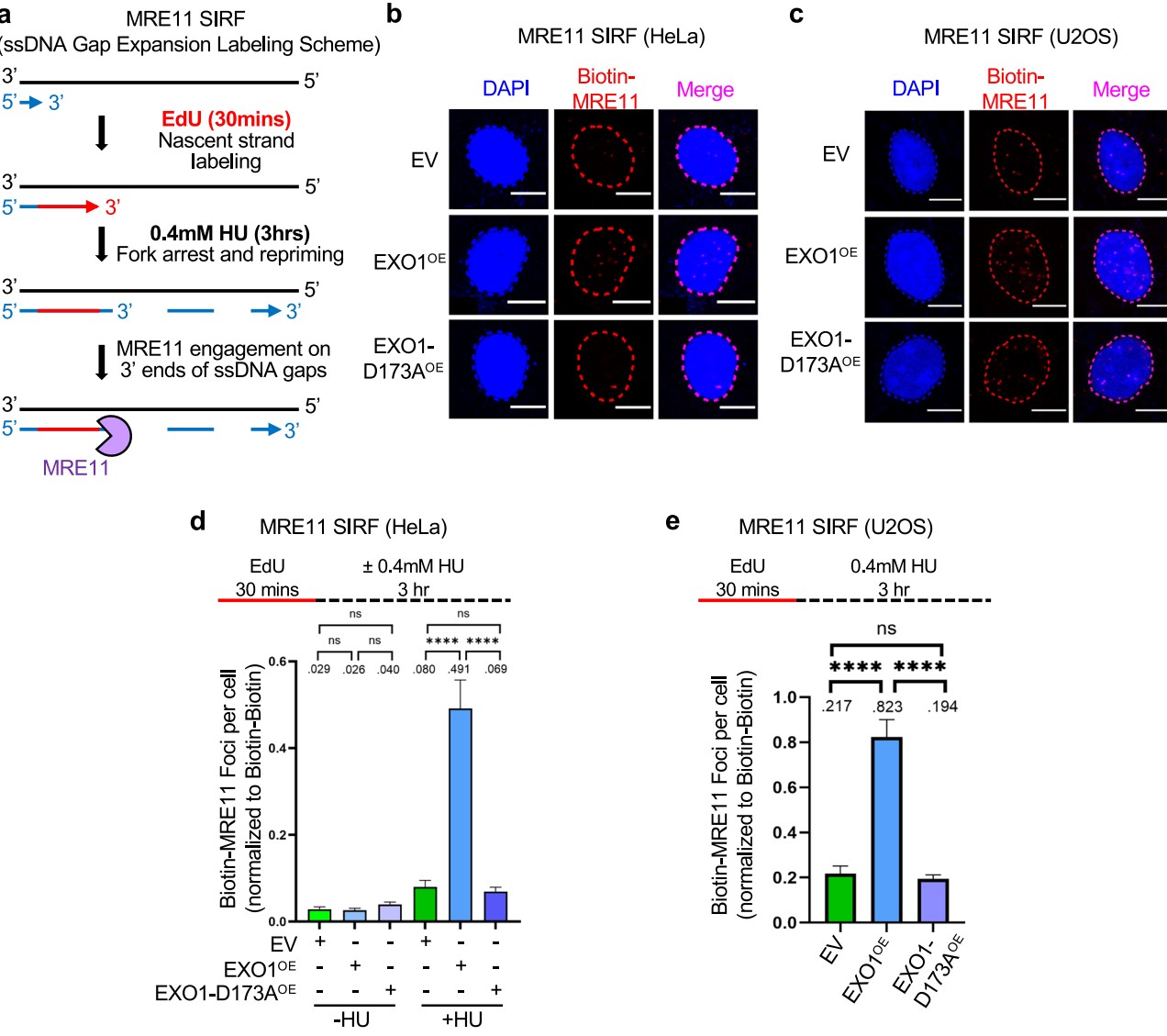

**Fig. 3 | EXO1 overexpression promotes MRE11 recruitment to nascent DNA for gap expansion. a–e** SIRF experiments showing that treatment with 0.4 mM HU induces binding of MRE11 to nascent DNA in EXO1-overexpressing cells HeLa (**b**, **d**) and U2OS (**c**, **e**), but not in cells overexpressing the D173A catalytic inactive mutant. The labeling scheme (**a**) is designed to capture MRE11 binding to the 3' ends of the gaps (for simplicity, only one strand, e.g. the leading strand, is shown in the schematic representation; EdU-labeled nascent DNA is indicated in red). Representative micrographs with scale bars representing 10 μm (**b**, **c**) and quantifications (**d**, **e**) are shown. At least 100 cells were quantified for each condition. Bars indicate the mean values, error bars represent standard errors of the mean, and asterisks indicate statistical significance (*t* test, two-tailed, unpaired). Schematic representations of the assay conditions are shown at the top. Source data are provided as a Source Data file.

dose (0.4 mM for 30 mins) of hydroxyurea (HU), which depletes nucleotide pools, reducing replication fork progression, and thus mimicking nucleotide deprivation which is observed in hyper-replicating cells during cancer initiation[58]. Under these conditions, we observed an increase in the alkaline comet tail moment in BrdU-labeled EXO1-overexpressing cells, but not in cells overexpressing the catalytic mutant, in both HeLa and U2OS cells (Fig. 1c–e).

We also induced replication stress by acute treatment (150 μM for 30 mins) with cisplatin, a chemotherapeutic agent which causes guanine adducts on DNA and thus interferes with DNA replication[59,60]. Similar to the HU experiments, we observed an increase in the alkaline comet tail moment in BrdU-labeled EXO1-overexpressing cells, but not in cells overexpressing the catalytic mutant, in both HeLa and U2OS cells (Fig. 1f, g). As a positive control, we employed cells overexpressing PRIMPOL, which have increased ssDNA levels due to increased repriming downstream of DNA lesions[8,11,40]. As we previously showed[40], PRIMPOL overexpression causes an increase in BrdU olive

tail moment. These findings suggest that, upon replication stress, EXO1 overexpression causes replication-associated DNA lesions, and its catalytic activity is required for this.

## EXO1 overexpression results in accumulation of ssDNA gaps in BRCA-proficient cells by promoting MRE11 recruitment to nascent DNA for gap expansion

We previously showed that EXO1 expands nascent strand ssDNA gaps formed upon replication stress in BRCA2-deficient cells[39]. Since the BrdU alkaline comet assay can detect ssDNA gaps, and the replication stress conditions we used for the experiments described above are known to cause ssDNA gaps[38,40,61], we tested if the BrdU alkaline comet signal may derive from increased gap accumulation. To this end, we measured ssDNA gaps using the S1 nuclease DNA fiber combing assay[62]. We incubated cells consecutively with thymidine analogs IdU and CldU and treated them with 0.4 mM HU during the CldU incubation. Treatment of purified DNA fibers with

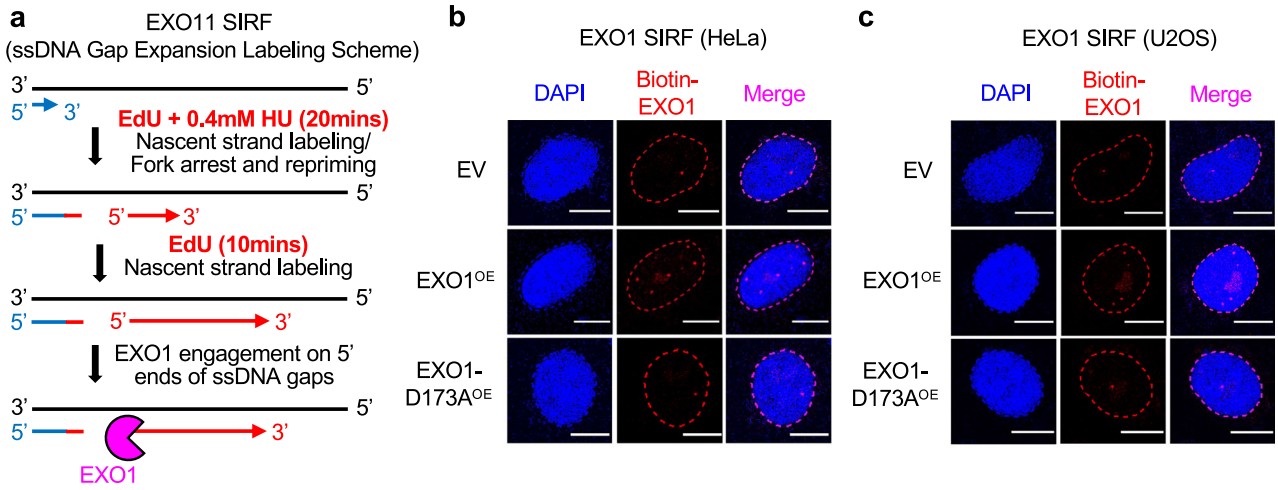

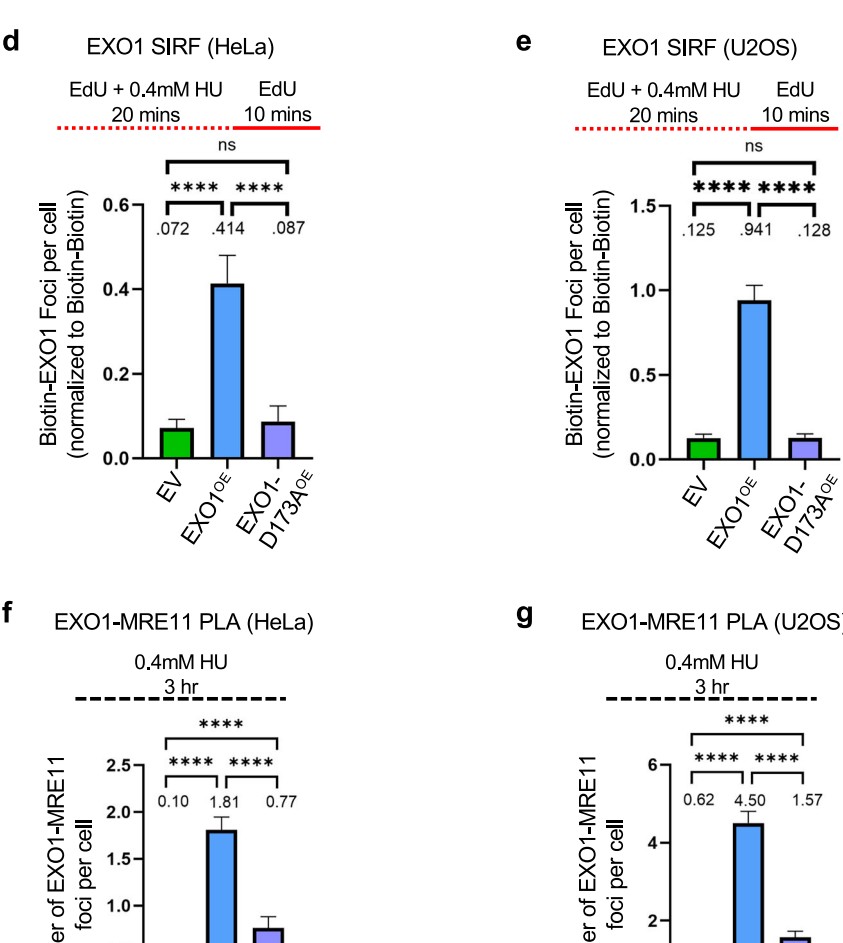

the ssDNA endonuclease S1 results in cleavage of DNA regions exposing ssDNA.

Using this assay, we observed ssDNA gap accumulation in HU-treated EXO1-overexpressing HeLa and U2OS cells, as evidenced by a decrease in CldU/IdU ratios in S1-treated cells (Fig. 2a, b). In contrast, ssDNA gaps were not detected in EV control cells or in cells over-expressing the D173A catalytic mutant. Similar findings were observed upon treatment with 150 μM cisplatin, in both HeLa and U2OS cells (Fig. 2c, d). These findings suggest that increased EXO1 levels are enough to cause ssDNA gap accumulation, even in cells with functional BRCA pathway.

We previously showed that EXO1 interacts with MRE11 to bidir-ectionally expand ssDNA gaps in BRCA-deficient cells[39,40]. Since we observed ssDNA gap accumulation in EXO1-overexpressing cells, we

**Fig. 4 | EXO1 localization to ssDNA gaps. a-e**. SIRF experiments showing that treatment with 0.4 mM HU induces binding of EXO1 to nascent DNA in EXO1-overexpressing cells HeLa (**b, d**) and U2OS (**c, e**), but not in cells overexpressing the D173A catalytic inactive mutant. The labeling scheme (**a**) is designed to capture EXO1 binding to the 5′ ends of the gaps (for simplicity, only one strand, e.g. the leading strand, is shown in the schematic representation; EdU-labeled nascent DNA is indicated in red). Representative micrographs with scale bars representing 10 μm (**b, c**) and quantifications (**d, e**) are shown. At least 100 cells were quantified for each condition. Bars indicate the mean values, error bars represent standard errors

of the mean, and asterisks indicate statistical significance (t-test, two-tailed, unpaired). Schematic representations of the assay conditions are shown at the top. **f, g**. PLA experiments showing that MRE11 and EXO1 co-localize in EXO1-overexpressing HeLa (**f**) and U2OS (**g**) cells, but not in cells overexpressing the D173A catalytic inactive mutant, upon treatment with 0.4 mM HU for 3 h. At least 100 cells were quantified for each condition. Bars indicate the mean values, error bars represent standard errors of the mean, and asterisks indicate statistical significance (t-test, two-tailed, unpaired). Source data are provided as a Source Data file.

employed the proximity ligation (PLA)-based SIRF (in situ quantification of proteins interactions at DNA replication forks) assay, which allows the quantification of specific protein recruitment to nascent DNA[63], to investigate if increased EXO1 levels are enough to recruit MRE11 to nascent DNA under replication stress conditions. We employed a SIRF labeling scheme designed to detect MRE11 engagement on gaps considering its 3′ −5′ directionality of exonuclease activity: we first labeled cells with EdU for 30 min, then removed it and added 0.4 mM HU for 3 h in order to measure MRE11 engagement on the 3′ ends of the gaps (Fig. 3a). We found that MRE11 is recruited to nascent DNA in EXO1-overexpressing cells but not in cells over-expressing the D173A catalytic mutant, in both HeLa and U2OS cells (Fig. 3b–e). As a control, in the absence of HU treatment, MRE11 was not recruited to nascent DNA (Fig. 3d).

We next investigated the localization of EXO1 itself. We employed a SIRF labeling scheme designed to detect EXO1 engagement on gaps considering its 5′−3′ exonuclease activity: we labeled cells with EdU for 30 min, with 0.4 mM HU added only during the first 20 min of EdU labeling, in order to measure EXO1 engagement on the 5′ ends of the gaps (Fig. 4a). In both HeLa and U2OS cells, we found increased EXO1 recruitment to nascent DNA in cells overexpressing wildtype EXO1, but not in those overexpressing the D173A mutant (Fig. 4b–e).

Since we previously showed that MRE11 and EXO1 co-localize in BRCA-deficient cells[39], we also tested if EXO1 overexpression causes an increase in the formation of EXO1-MRE11 complexes. Using PLA experiments we found that, in both HeLa and U2OS cells, MRE11 and EXO1 show increased co-localization upon treatment with 0.4 mM HU in EXO1-overexpressing cells, and this interaction is reduced in cells overexpressing the EXO1 D173A mutant (Fig. 4f, g). This suggests that increased EXO1 activity enhances the localization of MRE11 to nascent DNA in BRCA-proficient cells, and the co-localization of EXO1 and MRE11 at these structures.

## EXO1 overexpression promotes the degradation of reversed replication forks in BRCA-proficient cells

Upon prolonged fork arrest, replication forks reverse into a 4-way structure which exposes a DSB end. The BRCA pathway stabilizes the reversed fork[6,8,19–22]. We previously showed that EXO1 collaborates with MRE11 to catalyze the degradation of both strands of the DSB end of reversed replication forks in BRCA-deficient cells[26]. We thus tested if EXO1 overexpression can promote the degradation of stalled replication forks, even in BRCA-proficient cells. To induce replication fork reversal, we treated cells with high-dose (4 mM) HU which causes acute fork arrest[64]. We used the DNA fiber combing assay using a labeling scheme designed to investigate nascent strand degradation, namely consecutive incubation with thymidine analogs IdU and CldU for 30 mins each, followed by treatment with 4 mM HU for 4 h[64]. Nascent strand degradation is detected as a reduction in CldU/IdU ratios. We found that EXO1 overexpression, but not overexpression of the D173A mutant, resulted in nascent strand degradation, in both HeLa and U2OS cells (Fig. 5a, b). We also employed this assay to examine stalled fork degradation in BRCA-proficient breast cancer cell lines. We found that T47D cells, which showed the highest level of EXO1 expression in our hands, have increased replication fork degradation compared to MDA-MB-468

and MCF7 cells, which showed comparatively less EXO1 expression (Supplementary Fig. 1b, 3a–e).

We next tested if, similar to ssDNA gap expansion, MRE11 activity is also involved in the degradation of reversed forks in EXO1-overexpressing cells. To this end, we first employed SIRF experiments to investigate the engagement of MRE11 to reversed replication forks[21]. We labeled cells with EdU for 20 min and then treated them with 4 mM HU for 4 h to induce fork arrest and reversal (Fig. 5c). Under these conditions, we found increased MRE11 engagement on nascent DNA in EXO1-overexpressing cells but not in cells overexpressing the D173A catalytic mutant, in both HeLa and U2OS cells (Fig. 5d–g). Confirming that the MRE11 SIRF signal in EXO1-overexpressing cells reflects MRE11 engagement on reversed replication forks, this signal was abolished upon knockdown of the fork reversal translocase SMARCAL1, which is required for fork reversal[65] (Fig. 5h, Supplementary Fig. 4a).

The increased MRE11 binding to reversed forks in EXO1-overexpressing cells was surprising, considering that these cells are proficient for BRCA-mediated loading of RAD51 to these structures. Surprisingly, we found that EXO1-overexpressing cells show increased MRE11 SIRF signal compared to BRCA2-depleted cells (Fig. 5h), arguing that increased EXO1 activity promotes MRE11 engagement on reversed forks to a greater extent than loss of BRCA2-mediated RAD51 coating of reversed forks. Importantly, when we knocked down BRCA2 in EXO1-overexpressing cells, we found no additional increase in MRE11 loading, suggesting that EXO1 overexpression completely alleviates the fork protection activity of BRCA2.

In line with this, DNA fiber combing experiments showed that both inhibition of MRE11 exonuclease activity using the specific inhibitor mirin, as well as its siRNA-mediated depletion, suppress the fork degradation observed in EXO1-overexpressing HeLa and U2OS cells, indicating that MRE11 nuclease activity is critical for this degradation (Fig. 5i, Supplementary Fig. 4b–d). Confirming that this degradation occurs on reversed replication forks, knockdown of the fork reversal translocases ZRANB3 and SMARCAL1[65] also suppressed fork degradation in EXO1-overexpressing cells (Fig. 5i). Moreover, the levels of ZRANB3 and SMARCAL1, as well as those of PRIMPOL and MRE11 were not affected in EXO1-overexpressing cells compared to control cells (Supplementary Fig. 5a–d), ruling out an unspecific effect of EXO1 overexpression. These findings indicate that EXO1 overexpression causes degradation of reversed forks in BRCA-proficient cells, through engagement of MRE11.

## EXO1 overexpression causes DSB accumulation and chemotherapy sensitization

We previously showed that, in both BRCA-deficient and PRIMPOL-overexpressing cells, ssDNA gap expansion by MRE11 and EXO1 results in DSB formation[39,40]. Moreover, degradation of reversed forks may also promote DSB formation[36]. We thus investigated if EXO1 overexpression is enough to cause DSB accumulation upon replication stress, even in BRCA-proficient cells. We quantified DSBs using the neutral comet assay. Treatment with either 0.4 mM HU or 150 μM cisplatin for 2 h resulted in DSB accumulation in EXO1-overexpresing cells but not in cells overexpressing the D173A catalytic mutant, in both HeLa and U2OS cells (Fig. 6a–d). Moreover,

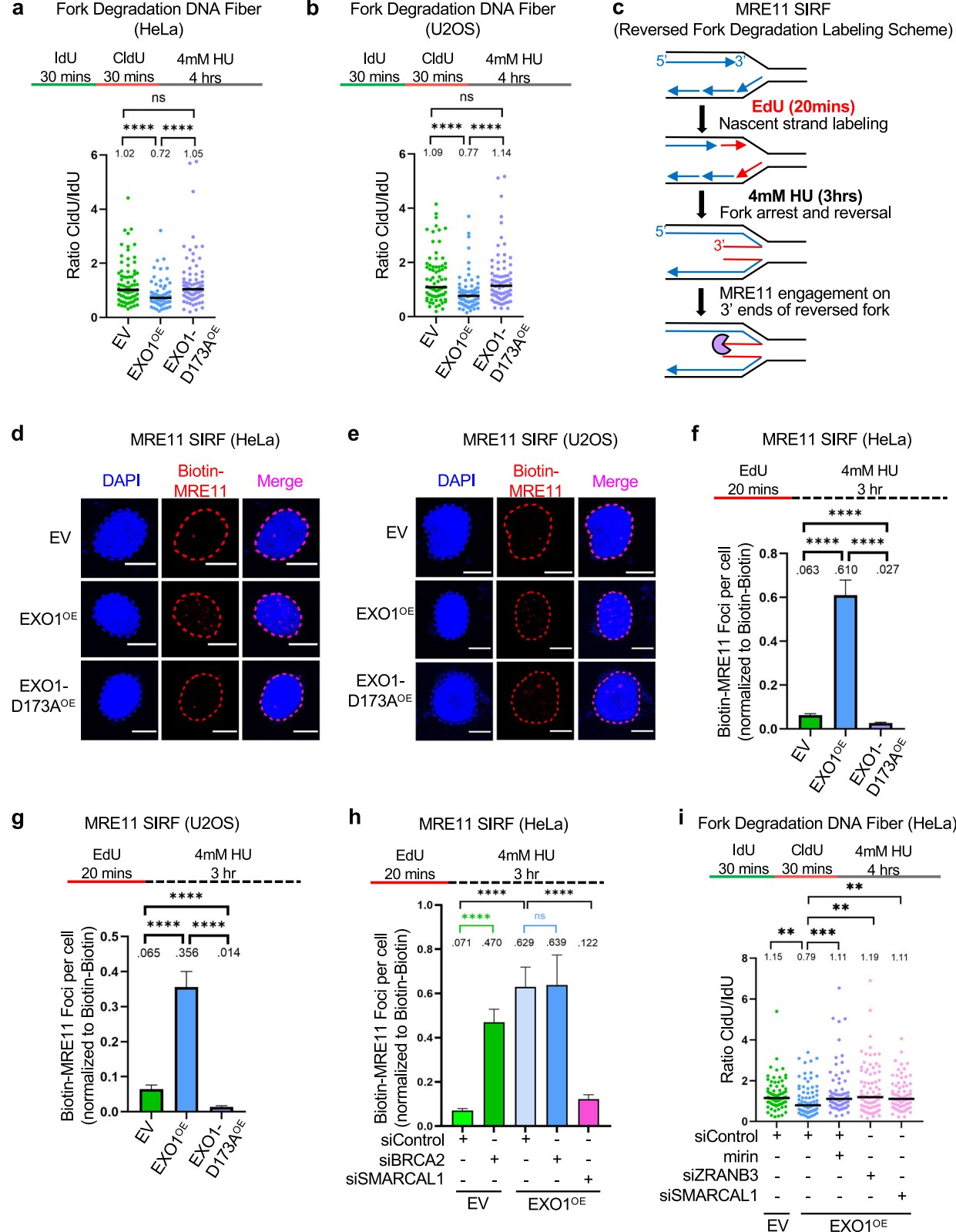

γH2AX foci were specifically increased upon treatment with 500 nM cisplatin for 20 h (Fig. 6e, f). These findings indicate that EXO1 overexpression causes DSB formation under replication stress conditions, even in BRCA-proficient cells.

Since the experiments presented above indicated that EXO1 overexpression causes DSB formation in response to cisplatin treatment, we measured if EXO1 overexpression increases the cellular sensitivity to cisplatin. We treated cells with 0.2 μM cisplatin for 10 days and observed that EXO1-overexpressing HeLa cells were more sensitive to the drug treatment compared to control cells (Fig. 7a). In contrast, overexpression of the D173A catalytic mutant did not affect cisplatin sensitivity compared to control cells. Increased sensitivity to DNA damaging agents, degradation of reversed replication forks, and accumulation of ssDNA gaps are hallmarks of BRCA pathway

**Fig. 5 | EXO1 overexpression promotes the degradation of reversed replication forks in BRCA-proficient cells.** DNA fiber combing assays showing that treatment with 4 mM HU for 4 h cause fork degradation in EXO1-overexpressing HeLa (**a**) and U2OS (**b**) cells, but not in cells overexpressing the D173A catalytic inactive mutant. The ratio of CldU to IdU tract lengths is presented, with the median values marked on the graphs and listed at the top. At least 75 tracts were quantified for each sample. Asterisks indicate statistical significance (Mann-Whitney, two-tailed). Schematic representations of the assay conditions are shown at the top. **c**–**g** SIRF experiments showing that treatment with 4 mM HU induces binding of MRE11 to nascent DNA in EXO1-overexpressing HeLa (**d, f**) and U2OS (**e, g**) cells, but not in cells overexpressing the D173A catalytic inactive mutant. The labeling scheme (**c**) is designed to capture MRE11 binding to the DSB end of the reversed replication fork (EdU-labeled nascent DNA is indicated in red). Representative micrographs with scale bars representing 10 μm (**d, e**) and quantifications (**f, g**) are shown. At least 100 cells were quantified for each condition. Bars indicate the mean values, error bars represent standard errors of the mean, and asterisks indicate statistical significance

(*t* test, two-tailed, unpaired). Schematic representations of the assay conditions are shown at the top. **h** SIRF experiment showing that MRE11 recruitment to nascent DNA induced by treatment with 4 mM HU in EXO1-overexpressing cells is not further enhanced by BRCA2 depletion, but is suppressed by depletion of SMARCAL1. At least 100 cells were quantified for each condition. Bars indicate the mean values, error bars represent standard errors of the mean, and asterisks indicate statistical significance (*t* test, two-tailed, unpaired). Schematic representations of the assay conditions are shown at the top. **i** DNA fiber combing assay showing that fork degradation induced by treatment with 4 mM HU for 4 h in EXO1-overexpressing HeLa cells is suppressed by MRE11 exonuclease inhibition using the specific inhibitor mirin, or by knockdown of the fork reversal translocases ZRANB3 and SMARCAL1. The ratio of CldU to IdU tract lengths is presented, with the median values marked on the graphs and listed at the top. At least 75 tracts were quantified for each sample. Asterisks indicate statistical significance (Mann–Whitney, two-tailed). Schematic representations of the assay conditions are shown at the top. Source data are provided as a Source Data file.

deficiency. BRCA-mutant cells are hypersensitive to PARP1 inhibitors. We thus tested the PARP1 inhibitor sensitivity of EXO1-overexpressing cells. We observed that HeLa cells overexpressing wildtype EXO1, but not cells overexpressing the EXO1 catalytic mutant D173A, are also sensitive to the PARP1 inhibitor olaparib (Fig. 7b, c). The olaparib sensitivity of EXO1-overexpressing cells was similar to that of BRCA2-knockout cells (Fig. 7c). Overall, these findings show that increased EXO1 activity sensitizes BRCA-proficient cells to cisplatin and PARP1 inhibitors.

## Discussion

In this work, we show that EXO1 overexpression causes nascent strand degradation upon replication stress, through both ssDNA gap expansion and reversed fork degradation, and this promotes genomic instability but also cisplatin sensitization in BRCA-proficient cells (Fig. 7d). Our previous work showed that EXO1 collaborates with MRE11 to degrade nascent DNA at both ssDNA gaps and reversed replication forks[26,39,40]. Since MRE11 and EXO1 have opposite directionalities, they can efficiently degrade nascent DNA on both strands. However, these studies were performed in BRCA-deficient cells, which are defective in the protection of reversed forks, as well as in HR-mediated gap filling. Here, we show that increasing EXO1 levels is enough to drive nascent strand degradation, at both ssDNA gaps and reversed forks, in BRCA-proficient cells.

Our results show that EXO1 overexpression bypasses the fork protection activity of the BRCA pathway, which loads RAD51 to protect against MRE11. Depletion of BRCA2 in EXO1-overexpressing cells did not cause an additional increase in MRE11 loading to nascent DNA. This epistatic interaction indicates that EXO1 overexpression alleviates the fork protection activity of BRCA2, arguing that the presence of RAD51 on reversed forks is not enough to block MRE11-mediated fork degradation. These results suggest that RAD51 loading on nascent DNA is not critical for fork protection, as previously thought, and instead argue that the BRCA pathway can promote fork protection in a RAD51-independent manner. Indeed, it was previously speculated that the BRCA proteins may be able to directly inhibit MRE11[11]. Perhaps, in BRCA-proficient cells, increased EXO1 activity and subsequent MRE11 recruitment is enough to bypass the inhibition of MRE11 by the BRCA pathway.

We also show that this EXO1 effect is mediated by its exonuclease catalytic activity, as overexpression of the D173A catalytic mutant does not cause nascent strand degradation and genomic instability. Interestingly, the catalytic mutant showed reduced co-localization with MRE11, perhaps suggesting that EXO1 activity is important to stabilize MRE11-EXO1 complexes on nascent DNA. Indeed, we show that both EXO1 and MRE11 are localized to ssDNA gaps in BRCA-proficient cells overexpressing wildtype EXO1, but not the D173A catalytic mutant.

This suggests that the catalytic activity of EXO1 is essential for MRE11-mediated gap expansion. BRCA-proficient cells can quickly fill ssDNA gaps by recombination-mediated gap filling. We speculate that increased EXO1 activity may interfere with this process, by expanding the gap and thus providing more opportunities for MRE11 to engage the other end of the gap and further drive bidirectional gap expansion, alleviating the ability of the BRCA pathway to fix the gap.

In line with previous studies[43–47], we show that EXO1 is overexpressed in cancers. Since genomic instability is an enabling characteristic of tumors, it is possible that EXO1 overexpression may be a driver of carcinogenesis in these cancers. On the other hand, it is possible that EXO1 amplification is a passenger event, since EXO1 is located on a chromosome 1 arm (1q) which has been shown to be amplified in many breast cancer samples. Passenger events have indeed been shown to contribute to cancer cell fitness[66–69]. Either way, our results suggests that EXO1 amplification may be relevant for cancer cells. EXO1 overexpressing cells show genomic instability under replication stress conditions, such as nucleotide deprivation. While the accepted paradigm is that DNA repair genes have tumor suppression activity by reducing genomic instability, our study suggests that DNA repair factors may potentially also promote cancer when their activity is unleashed. This reflects the complexity of DNA repair mechanisms, with multiple enzymes acting coordinately to fix DNA lesions in a process which often entails removing normal nucleotides adjacent to the lesion and highlights the need to tightly control the activities of these factors.

Finally, we show that EXO1 overexpression causes DSB formation in response to cisplatin, as well as chemosensitivity. Thus, our study suggests that EXO1 overexpression causes a "BRCAness" phenotype, characterized by genomic instability and hypersensitivity to DNA damaging chemotherapy[70]. Since increased EXO1 expression is found in a higher proportion of cancers than BRCA inactivation, this has important implications for both cancer etiology as a prevalent mechanism of genomic instability, as well as cancer treatment, suggesting that EXO1 could be a biomarker for the response to replication stress-inducing cancer therapy.

## Methods
### Cell culture and protein techniques
HeLa (ATCC CCL-2), U2OS (ATCC HTB-96), T47D (ATCC HTB-133), MDA-MB-231 (ATCC HTB-26), MCF7 (ATCC HTB-22), MDA-MB-468 (ATCC HTB-132), and MCF10A (ATCC CRL-10317) cells, obtained from ATCC, were grown in Dulbecco's modified Eagle's media (DMEM) supplemented with 15% FBS and penicillin/streptomycin. For overexpression of wildtype or D173A EXO1 variants, pLV[Exp]-Puro-CMV>hEXO1 lentiviral constructs (VectorBuilder) were used. Infected cells were selected by puromycin. PRIMPOL-overexpressing HeLa and U2OS cells were generated in our laboratory and previously

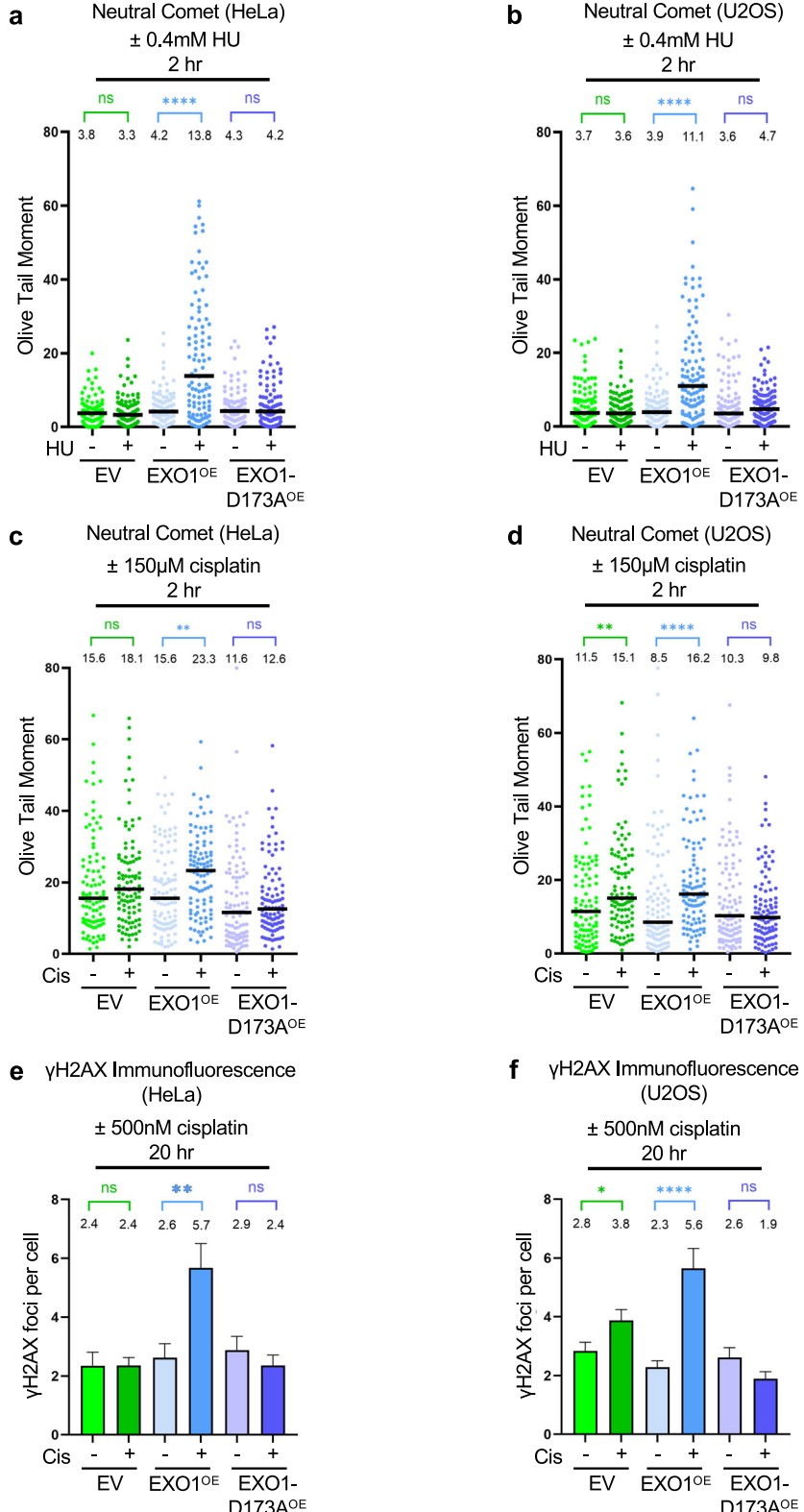

**Fig. 6 | EXO1 overexpression causes DSB accumulation.** Neutral comet assays showing that treatment with 0.4 mM HU (**a**, **b**) or 150 μM cisplatin (**c**, **d**) for 2 h causes accumulation of DSBs in EXO1-overexpressing HeLa (**a**, **c**) and U2OS (**b**, **d**) cells, but not in cells overexpressing the D173A catalytic inactive mutant. At least 100 comets were quantified for each sample. The median values are marked on the graph, and asterisks indicate statistical significance (Mann–Whitney, two-tailed).

γH2AX immunofluorescence showing that that treatment with 0.5 μM cisplatin for 20 h increases γH2AX foci in EXO1-overexpressing HeLa (**e**) and U2OS (**f**) cells, but not in cells overexpressing the D173A catalytic inactive mutant. At least 100 cells were quantified for each condition. Bars indicate the mean values, error bars represent standard errors of the mean, and asterisks indicate statistical significance (*t* test, two-tailed, unpaired). Source data are provided as a Source Data file.

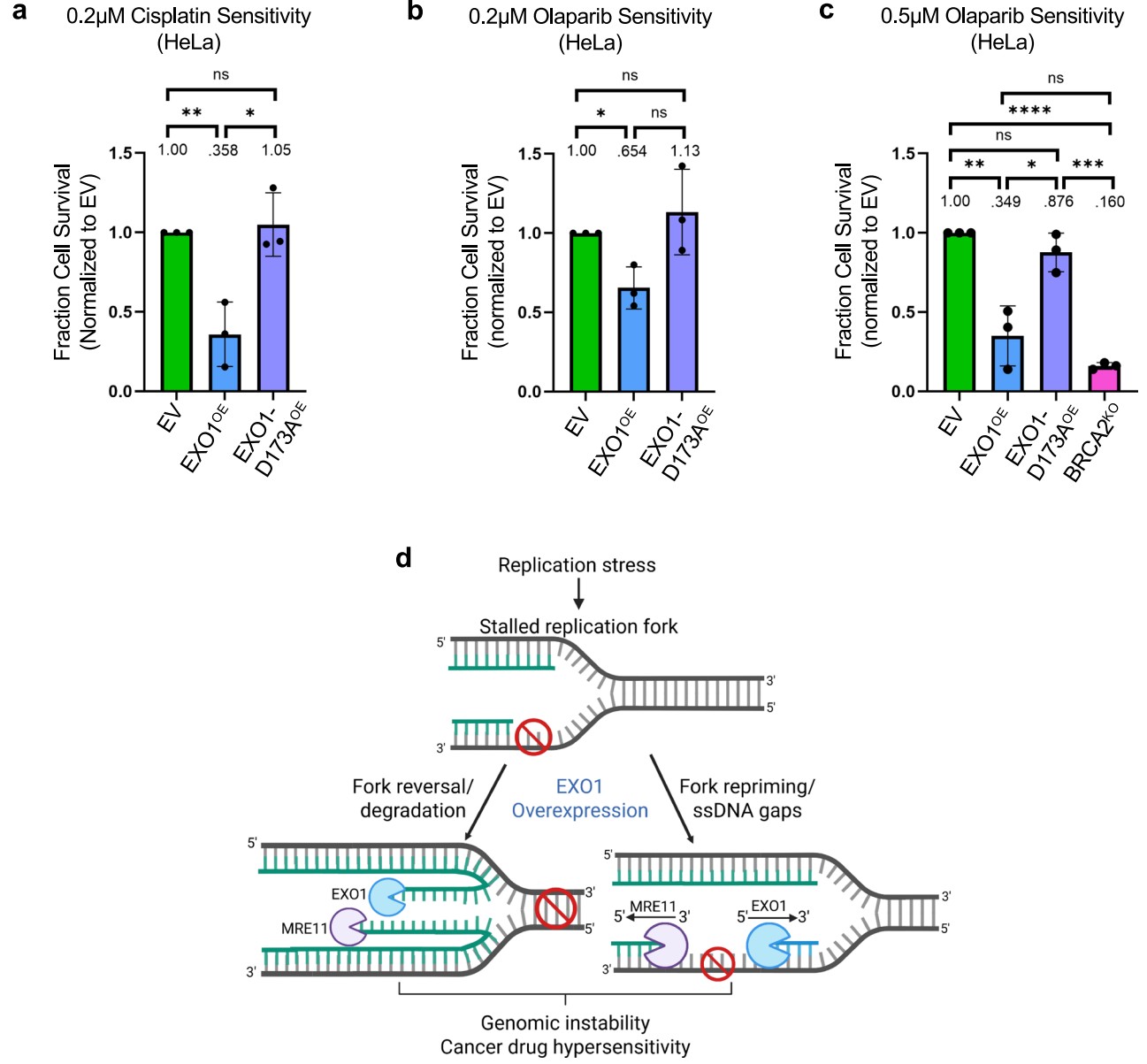

**Fig. 7 | EXO1 overexpression causes chemotherapy sensitization.** Cellular viability assays showing cisplatin (**a**) and olaparib (**b**, **c**) sensitivity of EXO1-overexpressing HeLa cells. Cells were grown in the presence of 0.2μM cisplatin (**a**), 0.2μM olaparib (**b**) or 0.5 μM olaparib (**c**) for 10 days. Cellular sensitivity was calculated compared to non-treated cells. Sensitivity of EXO1-overexpressing cells is shown as normalized to that of control (empty vector) cells. The average of three independent experiments, with standard deviations indicated as error bars, is shown. Asterisks indicate statistical significance (t-test, two-tailed, unpaired). **d** Schematic representation of the proposed model. Similar to BRCA deficiency, EXO1 overexpression in tumors causes genome instability through increased nascent strand degradation, both at ssDNA gaps and at reversed replication forks. This results in double strand break formation and chemosensitivity. Created in BioRender. Moldovan, G. (2026) https://BioRender.com/x4aglco. Source data are provided as a Source Data file.

described[40]. BRCA2-knockout HeLa cells were generated in our laboratory and previously described[71].

Gene knockdown was performed using Lipofectamine RNAiMAX. AllStars Negative Control siRNA (Qiagen 1027281) was used as control. The following oligonucleotide sequences (Stealth or SilencerSelect siRNA, ThermoFisher) were used:

ZRANB3: UGGCAAUGUAGUCUCUGCACCUAUA;
SMARCAL1: CACCCTTTGCTAACCCAACTCATAA;
BRCA2: AUUAGGAGAAGACAUCAGAAGCUUG;
MRE11: AAUAACUCGAGGCAGGUAUGUAAUG.

Denatured whole cell extracts were prepared by boiling cells in 100 mM Tris, 4% SDS, 0.5 M β-mercaptoethanol. Antibodies used for Western blot, at 1:500 dilution, were:

EXO1: Novus NBP2-16391;
GAPDH: Santa Cruz Biotechnology sc-47724;
SMARCAL1: Invitrogen PA5-54181;
ZRANB3: Novus NBP2-93301;
MRE11: Santa Cruz Biotechnology sc-135992;
PRIMPOL: Proteintech 29824-1-AP.

### Functional assays
Neutral and BrdU alkaline comet assays were performed as previously described[38] using the Comet Assay Kit (Trevigen, 4250-050). For the BrdU alkaline comet assay, cells were incubated with 100 μM BrdU as indicated. Chemical compounds (HU, cisplatin) were added according to the labeling schemes presented. Slides were stained with anti-BrdU

(BD 347580) antibodies and secondary AF568-conjugated antibodies (Invitrogen A-11031) and imaged on a Nikon microscope operating the NIS Elements V1.10.00 software. Olive tail moment was analyzed using CometScore 2.0. Immunofluorescence was performed as previously described[72] using a γH2AX antibody (MilliporeSigma JBW301). Slides were imaged on a confocal microscope (Leica SP5) and analyzed using ImageJ 1.53a software. For drug sensitivity assays, cells were grown for 10 days in the presence of 0.2μM cisplatin, 0.2μM olaparib, or 0.5μM olaparib, with splitting every 2-3 days. Cellular viability was determined using a NanoEntek EVE Automated Cell Counter.

## Proximity ligation-based assays

For PLA assays, cells were seeded into 8-chamber slides and 24 h later, treated as indicated in the respective labeling schemes. Cells were then permeabilized with 0.5% Triton for 10 min at 4 °C, washed with PBS, fixed at room temperature with 3% paraformaldehyde in PBS for 10 min, washed again in PBS, and then blocked in Duolink blocking solution (Millipore Sigma DUO82007) for 1 hr at 37 °C, and incubated overnight at 4 °C with primary antibodies. Antibodies used were: MRE11 (Genetex GTX70212) and EXO1 (Novus NBP2-16391). Samples were then subjected to a proximity ligation reaction using the Duolink kit (Millipore Sigma DUO92008) according to the manufacturer's instructions. Slides were imaged using a confocal microscope (Leica SP5) and images were analyzed using ImageJ 1.53a software.

For SIRF assays, cells were seeded into 8-chamber slides and 24 h later they were pulse-labeled with 50 μM EdU and treated with chemical compounds according to the labeling schemes presented. Cells were permeabilized with 0.5% Triton for 10 min at 4 °C, washed with PBS, fixed at room temperature with 3% paraformaldehyde in PBS for 10 min, washed again in PBS, and then blocked in 3% BSA in PBS for 30 min. Cells were then subjected to Click-iT reaction with biotin-azide using the Click-iT Cell Reaction Buffer Kit (ThermoFisher C10269) for 30 min and incubated overnight at 4 °C with primary antibodies diluted in PBS with 1% BSA. The primary antibodies used were: Biotin (mouse: Jackson ImmunoResearch 200-002-211; rabbit: Bethyl Laboratories A150-109A); MRE11 (GeneTex GTX70212); EXO1 (Santa Cruz Biotechnology sc-56092). Next, samples were subjected to a proximity ligation reaction using the Duolink kit (MilliporeSigma DUO92008) according to the manufacturer's instructions. Slides were imaged using a confocal microscope (Leica SP5) and images were analyzed using ImageJ 1.53a software. To account for variation in EdU uptake between samples, for each sample, the number of protein-biotin foci were normalized to the average number of biotin-biotin foci for that respective sample. The scale bars for the SIRF micrographs shown represent 10 μm.

## DNA fiber combing assays

Cells were incubated with 100 μM IdU and 100 μM CldU as indicated. Chemical compounds were added according to the labeling schemes presented. Next, cells were collected and processed using the Fiber-Prep kit (Genomic Vision EXT-001) according to the manufacturer's instructions. Samples were added to combing reservoirs containing MES solution (2-(N-morpholino) ethanesulfonic acid) and DNA molecules were stretched onto coverslips (Genomic Vision COV-002-RUO) using the FiberComb Molecular Combing instrument (Genomic Vision MCS-001). For S1 nuclease assays, MES solution was supplemented with 1 mM zinc acetate and either 40 U/mL S1 nuclease (ThermoFisher 18001016) or S1 nuclease dilution buffer as control, and incubated for 30 min at room temperature. Slides were then stained with antibodies detecting CldU (Abcam ab6326) and IdU (BD 347580) and incubated with secondary AlexaFluor 488 (Abcam ab150117) and Cy5 (Abcam ab6565) conjugated antibodies. Finally, the cells were mounted onto coverslips and imaged using a confocal microscope (Leica SP5) and analyzed using LASX 3.5.7.23225 software.

## Statistics analyses

For immunofluorescence, SIRF, and PLA assays, as well as cellular viability assays the $t$ test (two-tailed, unpaired) was used. For DNA fiber assays and comet assays, the Mann–Whitney statistical test (two-tailed) was performed. Western blot experiments were reproduced at least two times. Unless otherwise noted, schematic models were made with Microsoft PowerPoint v16.103. Statistical analyses were performed using GraphPad Prism 10 and Microsoft Excel v16.103. Statistical significance is indicated for each graph (ns = not significant, for $p > 0.05$; * for $p \leq 0.05$; ** for $p \leq 0.01$; *** for $p \leq 0.001$, **** for $p \leq 0.0001$).

## Reporting summary

Further information on research design is available in the Nature Portfolio Reporting Summary linked to this article.

## Data availability

All data supporting the findings of this study are available within the paper and its Supplementary Information. Source data are provided with this paper.

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

## Acknowledgements

We would like to thank Dr. Clare Sample and Cole Burgess for technical support and advice, as well as the Penn State College of Medicine Advanced Light Microscopy (RRID:SCR-022526) and Flow Cytometry (RRID:SCR_021134) core facilities. This work was supported by: NIH F31CA294862 (to AN), NIH R01ES026184 and NIH R01GM134681 (to GLM), NIH R01CA244417 (to CMN), as well as the Four Diamonds Transformative Patient-Oriented Cancer Research Project 4D01_2024_1002 (to GLM and CMN). The content is solely the responsibility of the authors and does not necessarily represent the official views of Four Diamonds. This manuscript is the result of funding in whole or in part by the National Institutes of Health (NIH). It is subject to the NIH Public Access Policy. Through acceptance of this federal funding, NIH has been given a right to make this manuscript publicly available in PubMed Central upon the Official Date of Publication, as defined by NIH.

## Author contributions

A.N., C.M.N., and G.L.M. designed the experiments; A.N. and C.M.N. performed the experiments; A.N. and G.L.M. wrote the manuscript.

## Competing interests

The authors declare no competing interests.
