## [Transparent Peer Review file · Nature Communications]

The nuclease EXO1 promotes genomic instability by degrading nascent DNA in BRCA-proficient cells

Corresponding Author: Dr George-Lucian Moldovan

Version 0:

Reviewer comments:

Reviewer #1

(Remarks to the Author)

The manuscript examines the role of EXO1 overexpression in inducing genomic instability in BRCA positive breast cancers. The authors use the cBioPortal to interrogate TCGA datasets and find many cancers with amplifications and higher expression levels of EXO1, including around 20% of breast cancers. They then established HeLa and U2OS based EXO1 over-expressing cell lines to examine the phenotypic effects. They find that EXO1 overexpression leads to nascent strand degradation at ssDNA gaps and reversed forks using S1 nuclease fork assays and BrdU comet assays. They show using SIRF/PLA that active EXO1 localizes to nascent DNA in proximity to MRE11 and find that EXO1 OE leads to nascent DNA degradation at forks in HU treated cells that can be prevented by MRE11 inhibition or depletion of fork reversal proteins like SMARCAL1 and ZRANB3. Finally, they demonstrate using neutral comet and IF that EXO1 OE leads to DSBs and higher sensitivity to cisplatin, that they conclude may explain the differences in survival seen in TCGA data. Overall, they find that elevating EXO1 expression leads to nascent strand degradation in BRCA proficient cancers, in addition to its previously known activity in BRCA deficient cancers.

The group has a strong track record performing the types of assays used in the paper and the experimental data is for the most part well-presented and convincing. With this, they show that EXO1 overexpression via lentivirus causes a number of phenotypes similar to its previously established role in BRCA deficient settings in multiple cell lines. My primary issue with the paper is the clinical data analysis and its interpretation and whether the OE system is truly modeling what they imply. The overexpression system is based on the finding that EXO1 is frequently amplified/upregulated in breast cancer. This occurs primarily in the context of a large 1q43 amplification that contains a large number of other genes. Whether EXO1 protein levels are in fact elevated and if so, to an extent that is similar to what they assay experimentally in their lentiviral system, remains an open question. I therefore have some doubts about the the relevance of the experimental system and I have very strong reservations about their bioinformatic analysis and its interpretation. I have separated my comments into those related to bioinformatics and experiments.

Major concerns

Bioinformatics

1. The authors stated: These findings argue that, despite the known role of EXO1 in DNA repair, increased EXO1 activity, rather than loss of EXO1 function, is associated with carcinogenesis. And later they state: Overall, these findings argue that increased EXO1 activity, rather than loss of EXO1 function, is associated with carcinogenesis, through increased genomic instability. This is a massive overinterpretation of the data. First, there is no indication in any sample that EXO1 activity, let alone its protein levels, are elevated. This could be addressed in tissue arrays by IHC or other methods but is not. Thus, this is an assumption based on mRNA or copy number. While it is possible that high EXO1 is more oncogenic than its loss, this data allows only a speculative point, as there is no evidence that amplified/overexpressed EXO1 is associated with clinical outcome and no evidence that it is responsible for the elevated genome instability. I make this point not to discount the experimental data in the paper, only that I think that this clinical data, that is very superficially analyzed, is being overinterpreted to make general statements that are unsupported by the data they are describing. If one wants to make prognostic claims about clinical data, at the very least, multivariate analyses on annotated clinical datasets should be used to calculate hazard ratios. BRCA deficient samples should also be separated from those datasets and the correlations with numerous other genes in the proposed pathway could also be assessed.
2. Related to this point, a major confounding factor in the types of univariate analyses produced by cBioPortal, and many

other online platforms, is proliferation. EXO1 is an E2F target and its expression highly correlated with a number of proliferation related genes (CHEK1, FOXM1, CCNA2 etc). Proliferation status often correlates with clinical outcomes, thus whether EXO1 per se is driving any clinical outcomes remains unclear, and is in fact argued against with the data used in Figure 7 (see next point on this). This, and other clinical parameters, need to be considered in survival data analysis using multiple variables

3. The authors stated: We found that breast cancer patients with altered EXO1 in tumors (which, as described above, represents amplification of EXO1 expression in the vast majority of cases) tend to have better survival than those with no EXO1 alterations (Fig. 7C,D). This is not what the data shows. First, this is a univariate analysis that is fine for hypothesis generation but not suited to interpret clinical data. Second, the p-values clearly indicate there is no significant difference in survival. The differences that the authors are referring to are a handful of the samples (in one case 4 patients of >1000). This in no way justifies a statement like the one made.

4. The authors stated: This is in line with the increased sensitivity of EXO1-overexpressing cells to cisplatin and suggests that EXO1 overexpression in tumors sensitizes them to cisplatin chemotherapy. Aside from the fact that these graphs show no difference in survival, how many of these patients received cisplatin?

5. The EXO1 amplifications that are apparent in the dataset used are also seen in other well curated cohorts like METABRIC. These do not represent EXO1 amplification in isolation but a very large duplication of 1q43 that contains a number of other genes that may also influence oncogenesis. If you run the same analyses with any of these genes (for example MROH9, SDHC, CHRM3, RGS7) you get precisely the same outcome- that there is no statistical difference in survival and there is a correlation with genome alterations. While the experimental data would lead to a reasonable hypothesis that EXO1 amp may play a role in genome alterations, it does not account for all of the other alterations happening with EXO1 amp in any way.

6. The authors stated: EXO1 is overexpressed in a significant proportion of cancers, including in over 20% of breast tumors. First, the graph referred to in Figure 1A is based on a specific dataset and cannot be generalized to all breast cancer. One can look at this in cBioPortal across many different BC datasets and see variations, as each one has its own biases in tumor subtypes and other factors. Second, it is not actually an accurate description of the cohort shown, as amplification is only shown in just over 5% of the Breast Cancer cohort (the actual cohort name should be used for proper reference), while the rest is mRNA high.

7. The phrase "all breast cancers" is used repeatedly in Figure 1 while describing data related to a single cohort. The proportion of basal cancers with elevated EXO1 is higher in many of the breast cohorts, however it is worth noting there are a lot more of other subtypes that have EXO1 high expression or amplifications. In fact, the p-value for survival is lower in some of these cohorts with fewer basal cancers (ex. Metabric), calling into question the relevance to TNBC. However, I still maintain that this type of 1:1 correlation is not a serious way to establish what the authors wish to conclude. If you use another platform that conducts multivariate analysis on the same dataset, the answer is in fact the opposite (ex MammoOnc-DB), where you see statistically worse survival in the high EXO1 cohort. I would highly recommend the authors consult a statistician to do this analysis in a more rigorous way across multiple datasets to make conclusions better supported by the clinical data. I would also point out previous data (I am not vouching for its accuracy, only its existence) that proposes that 1q amps that contain EXO1 are associated with poor prognosis in breast cancer (<https://doi.org/10.1371/journal.pone.0077553>).

Experimental data

1. For Figure 2, it would be nice to see some representative data in the figure rather than just the quantification.

2. All the experiments have been performed with EXO1 over-expression (WT/catalytically dead) in Hela/U2OS cells. As EXO1 activity is shown to be dependent on MRE11, PRIMPOL, SMARCA1 and ZRANB3, it would be reassuring to see that their relative levels are similar in the parental, WT and catalytically dead EXO1 lines.

3. The authors stated: This suggests that increased EXO1 levels results in recruitment of MRE11 to nascent DNA under replication stress through MRE11-EXO1 complex formation. Prior literature showed that the D173A mutant of EXO1 is catalytically dead, however it is capable of binding DNA as well as the WT, as shown previously by multiple groups (PMIDs:11842105, 40319035, 23178594) and it retains the PCNA binding PIP box (23939618). Has the localization of the WT and dead mutant been examined? It would seem possible that it is not simply levels, but levels and its activity that are needed.

4. Additionally, the authors argue that "increased EXO1 levels result in recruitment of MRE11 to nascent DNA under replication stress through MRE11-EXO1 complex formation". The MRN complex can bind DNA lesions in the absence of EXO1, so what more specifically are the authors proposing as the mechanism here? In the case of gaps, why is EXO1 needed for MRE11 to act in the other direction? While I would expect the resulting ssDNA to be shorter, I do not understand why MRN would not be able to act on these substrates without EXO1.

5. The authors stated: Thus, our study suggests that EXO1 overexpression causes a "BRCAness" phenotype, characterized by genomic instability and hypersensitivity to genotoxic chemotherapy. Are cells sensitive to PARP inhibitors?

Reviewer #2

(Remarks to the Author)

Reviewer #3

(Remarks to the Author)

The study investigates the role of the nuclease EXO1 in promoting genomic instability by degrading nascent DNA in cells that are proficient in the BRCA pathway. The researchers found that EXO1 is overexpressed in various cancers, leading to

increased degradation of nascent DNA and double-strand break (DSB) formation, even in the presence of functional BRCA genes. EXO1 is overexpressed (EXO1 OE) in a significant proportion of cancers, including breast, hepatocellular, skin, testicular, and cervical cancers. This overexpression is associated with increased genomic instability.

EXO1 OE in BRCA-proficient cells leads to replication-associated DNA lesions upon replication stress. This is evidenced by increased single-strand DNA (ssDNA) gaps and reversed replication forks. EXO1 OE promotes the recruitment of MRE11 to nascent DNA, facilitating the degradation of ssDNA gaps and reversed forks. This interaction is critical for the observed genomic instability. The degradation of nascent DNA by EXO1 results in increased DSB formation. This leads to hypersensitivity to genotoxic agents like cisplatin. EXO1 OE sensitizes cells to chemotherapy, particularly in triple-negative breast cancer, which lacks hormone receptors and is typically treated with chemotherapy.

Overall, the findings highlight the dual role of DNA repair enzymes like EXO1 in maintaining genomic stability under normal conditions and promoting genomic instability when overexpressed. This manuscript is a good contender for Nature Communications.

Figure 2 : The alkaline comet assay also detects dsDNA breaks, wouldn't their assay then also detect replication-associated dsDNA breaks ? (like fork collapse) or do they consider these breaks negligible, also because of the low dose of HU ?

Figures 2C and 2D indicate that the EXO1 D173A mutant exhibits lower activity in U2OS cells compared to HeLa cells. This suggests that the role or recruitment/localization dynamics of EXO1 D173A may vary depending on the cellular context, highlighting potential cell line-specific differences in how the mutant functions. Has this been addressed experimentally ?

The authors state that "MRE11 is recruited to nascent DNA in cells overexpressing EXO1 but not in cells overexpressing the catalytic mutant D173A, in both HeLa and U2OS cells." However, in Figure 4D, no significant difference in MRE11 recruitment is observed between EXO1 overexpression and EXO1 D173A overexpression in HeLa cells. Notably, there is also no significant difference between the empty vector (EV) and EXO1 D173A conditions, raising concerns about the robustness of this conclusion in the HeLa context.

Figure 4D : The error bar in the EXO1 D173A condition is very high. Could the authors show individual N (like on Figures 7A & B) ? It appears that the EXO1 mutant is still able to recruit Mre11 to some extent. Could the authors comment on this?

Figures 5C–G: How can the authors be certain that the SIRF signal they observe reflects degradation at inverted replication forks, especially without colocalization data using markers such as CtIP or MRE11?

Figure 5H: To strengthen the conclusion that degradation occurs at inverted forks via MRE11 activity, the authors should consider including an siMRE11 condition and assess for potential epistasis by performing a double siRNA knockdown.

The authors should compare the levels of EXO1 Overexpression in a panel of cancer cell lines to those in their HeLa and U2OS cells: are they comparable?

The most impactful conclusion of the paper is that EXO1-overexpressing cells exhibit a BRCAness phenotype in terms of sensitivity to DNA damage agents; however, the authors don't thoroughly explore this aspect. Figures 7C and D only bring circumstantial evidence that EXO1-overexpressing cancers are more sensitive to cisplatin. I think Figure 7 (and the paper) would be greatly improved if the authors could examine cell survival following cisplatin treatment in a panel of cancer cell lines that overexpress or do not overexpress EXO1.

Additionally, a major point lacking in the discussion is that many publications have shown that EXO1 over-expression is associated with a poor prognosis and cancer progression, due to the increase in genome instability.

Reviewer #4

(Remarks to the Author)

EXO1 is an exonuclease with important roles in long-range resection at sites of replication stress or DNA double-strand breaks, generating single-stranded DNA that can then be engaged by homologous recombination repair or be subject to further nuclease attack, potentially leading to more damage.

The study starts off with showing that EXO1 mRNA levels are increased in cancer samples, such as from breast and cervical cancer. This finding is not new as it has been reported before by several groups (PMID: 24147022, PMID: 33623391, PMID: 39718579, PMID: 33569437). It is not investigated whether EXO1 protein levels are also increased in cancer, which has been attempted by others using breast cancer cell lines (PMID: 24147022).

The further premise of the study is that increased EXO1 expression in cancers could cause genome instability phenotypes that resemble those caused by loss of DNA repair factors such as BRCA1. The study goes on to show that ectopic EXO1 protein overexpression in model cancer cell lines causes phenotypes consistent with increased proximity of MRE11 nuclease to EXO1 and nascent DNA, increased resection at nascent DNA and increased DNA double strand break formation – if these cells are treated with reagents that induce replication stress. EXO1-overexpressing cells are potentially more sensitive to cisplatin, although the evidence for this is limited. The conclusion is that EXO1 overexpression might cause a BRCAness-like phenotype that could sensitise cancers to chemotherapy.

The experiments are of good quality and nicely controlled by using the catalytically inactive (nuclease-deficient) EXO1 D173A mutant. This shows that most phenotypes are caused by the EXO1 nuclease catalytic activity rather than overexpression per se. EXO1 D173A is still able to bind DNA and interact with other DNA repair factors, which helps explain why nuclease-independent effects are sometimes seen such as in Figure 3A or Figure 4D, F.

The mechanistic findings are generally convincing and useful for the field, but are limited in breadth, using a small number of assays that have not evolved much from the author's previous publications on EXO1 and MRE11. The experiments show that increased EXO1 activity causes certain DNA replication and DNA damage phenotypes that would be expected if high EXO1 activity is detrimental, but do not provide much new insight into EXO1 function. The study does not directly measure effects on genomic instability or whether EXO1 might be an oncogene. The cancer data analysis is too rudimentary to add anything to the already existing literature.

Specific comments:

Top of page 5: The results section starts with: "To investigate the relevance of EXO1-mediated genome instability in cancer, (...)". This pre-supposes the findings of the study and does not explain why EXO1 expression levels in cancer were investigated to begin with. The authors could justify this much better. EXO1 is on located chromosome 1q, which is frequently amplified in breast cancer, and several aspects of high EXO1 expression in cancer have been characterised in previous studies (see above). This literature should be considered and cited.

Bottom of page 6: "However, we found that EXO1 expression positively correlates with genome alterations (Fig. 1D), suggesting that EXO1 promotes chromosomal structural instability (...)" Correlation is not causation. High EXO1 expression could be a consequence of chromosomal instability.

Bottom of page 7: "Overall, these findings argue that increased EXO1 activity, rather than loss of EXO1 function, is associated with carcinogenesis, through increased genomic instability." See previous comment. It seems fair to say that increased EXO1 expression is associated with cancers with high genomic instability. It is not certain that EXO1 is highly expressed during cancer development or that it increases genomic instability.

Figure 4: The authors should elaborate more on why they think EXO1 would recruit MRE11 to nascent DNA. The labelling scheme in Figure 4A is not very convincing as a gap-specific assay. Data in Figure 3 support that gaps may be formed in the DNA that is synthesised during the treatment with 0.4 mM HU. But for the experiments in Figure 4A-E, nascent DNA is labelled in absence of HU and 0.4 mM HU is added afterwards, providing less opportunity for the MRE11 that co-localises with EdU to be at gaps. It would be helpful to show a SIRF experiment without any HU treatment to see the baseline in unchallenged cells and to support the conclusion that the effect in Figures 4D,E and 5F,G shows MRE11 recruitment specifically to gaps and arrested forks, respectively.

Figure 7A, B: The link between the molecular findings in previous figures and cisplatin sensitivity would be more convincing if the sensitivity of EXO1 OE was normalised to the sensitivity of EXO1 D173A OE.

Materials and methods: Much more detail should be given on how the TCGA dataset analysis was performed and which datasets and search parameters used to generate Figure 1A and Figure 7C,D.

Version 1:

Reviewer comments:

Reviewer #1

(Remarks to the Author)

The authors have addressed many of the reviewer comments and included new data that helps support their original model and add additional depth. This included demonstrating that EXO1 activity is required for its retention, that its OE is epistatic with BRCA2 loss, that MRE11 is binding reversed forks with new SIRF data and that OE is roughly similar to some BC cell lines that are widely used. The overinterpretation of superficially analyzed clinical data has been toned down, and a subset has been moved to the supplement. My biggest reservation is that experiments are carried out for the most part in a single cell line background that is not a BC and has its own host of other genetic abnormalities, making some of the more generalized interpretations proposed difficult to support. While some data from cell lines expressing different levels of EXO1 is presented as further proof, this data is not particularly compelling.

Specific comments-

1. Referring to this comment in the rebuttal- "by showing that breast cancer cell lines with increased EXO1 expression have increased nascent strand degradation compared to cell lines with lower EXO1 expression levels – new Supplementary Figures S1C, S3A, S3B)." The authors argue that they have demonstrated that nascent strand degradation is elevated in T47D that are high EXO1 expressors of the lines examined. The data in S3A and S3B does not appear to support this. In S3A there is a minor difference, but not in S3B, and no statistics employed anywhere here. MDA-MB-468 cells that potentially have more than two-fold OE of EXO1 and no effect on nascent strand degradation, making this data unsupportive of authors claims.
2. The authors report a tissue microarray using EXO1 IF to determine levels. No primary data is provided to evaluate and no

indication of what controls were carried out to ensure that the EXO1 antibody was on target are provided. The methods provided state that 2 different primary EXO1 antibodies were used. Please clarify these points.

3. In referring to Figure S2D as measuring genomic instability, the authors should clarify exactly what types of lesions are measured in this assessment, as it is relevant to the potential mechanism. Do these types of aberrations make sense with the proposed elevation in nascent strand resection based on other examples?

4. In S2E-G, to me the most outstanding confounders are proliferation and the other genes amplified on 1q with EXO1. Do all 1q genes correlate with higher genome instability by this measure? Does proliferation? Looking quickly at CHK1 or KI67, it would appear so. The same applies to S2A, is this at all specific to EXO1 as is implied?

5. In the first sentence of the discussion, and several other places, it is stated that EXO1 OE causes nascent strand degradation. However, in the absence of RS, there is no indication of elevated breaks or damage, so it would be worth clarifying these statements to some extent in that they are dependent on fork stalling, as nicely shown by a number of data points. I am not asserting that the authors are trying to misrepresent anything, I just think it is not always clearly stated. Particularly, considering that all the experiments are done biologically unrelated cell types and the BC cell lines seem to have no significant effect of EXO1 OE.

6. I think it is a big leap to go from a HeLa cell model to proposing EXO1 as a biomarker generally for chemotoxic therapy. Is there any indication that it would outperform a proliferative marker for example? You would expect the same effects with a wide range of DNA damage types? This could be stated more specifically to align with the data.

7. The authors stated: Since increased EXO1 expression is found in a higher proportion of cancers than BRCA inactivation, this has important implications for both cancer etiology as a prevalent mechanism of genomic instability, as well as cancer treatment, suggesting that EXO1 could be a biomarker for the response to genotoxic chemotherapy. Relevant to this, the authors do show new data indicating that there is PARPi sensitivity in the OE HeLa cells. However, this data is somewhat weak as a single dose is used and not compared to BRCA mutations (including epistasis) in this background.

8. The EXO1 SIRF experiments are performed with treatment of HU for just 20 mins (Fig 4 A-E) while the EXO1-MRE11 PLA has been done after 4hrs of HU. Did the authors have any rationale for selecting the two different time points, did authors test earlier time points for EXO1-MRE11 PLA? If so, mentioning that would be useful to further substantiate that EXO1 activity triggers MRE11 recruitment.

Reviewer #2

(Remarks to the Author)

Reviewer #3

(Remarks to the Author)

The authors have satisfactorily addressed all my comments, and the manuscript is now acceptable for publication.

Reviewer #4

(Remarks to the Author)

The authors have addressed my comments to my satisfaction.

Version 2:

Reviewer comments:

Reviewer #1

(Remarks to the Author)

The authors have addressed the majority of my concerns and the expanded data in Figure 7 and Figure S3 add more confidence to the conclusions. I remain concerned about the data in S1B due and have to reiterate one of my primary concerns that was ignored in the response- no primary data is provided to evaluate.

2. The authors report a tissue microarray using EXO1 IF to determine levels. No primary data is provided to evaluate and no indication of what controls were carried out to ensure that the EXO1 antibody was on target are provided. The methods provided state that 2 different primary EXO1 antibodies were used. Please clarify these points.

We apologize but we do not understand why the reviewer states that "2 different primary EXO1 antibodies were used". We clearly state that the EXO1 primary antibody Santa Cruz Biotechnology sc-56092 was used.

I think due to the wording, I thought the Novus product was also an antibody, thanks for clarifying that.

The experiment presented was performed on a commercially available tissue microarray slide (Novus NBP2-30212). We were not able to perform control experiments (eg. no primary

antibody) since all samples were on the same slide, so they could not be differentially treated. We did perform control single antibody experiments but those were performed on a different tissue sample that was available to us. We observed minimal background signal in those control experiments. We are presenting below a representative micrograph showing this. We prefer to not include this in the manuscript, as the sample used for the test is different than the breast cancer microarray samples quantified in the figure. In addition, to this control, we would like to point out that the observed difference in EXO1 expression between breast cancer and normal tissue we observed, clearly indicates the specificity of the antibody.

The SC antibody listed seems to be discontinued, but archived data sheets do not mention IHC as a tested application or show examples of IF or tissue staining. If a tissue microarray was stained and scored, that "clearly indicates the specificity of the antibody", I think providing some example images of what was actually scored is a very reasonable request that was ignored in the author response. I also think that the case made for the control data is reasonable and do not understand why it would not be shown, absent any other more rigorous controls (we typically prepare FFPE blocks using KO cell line pellets for example).

There is currently a lot of clinical interest in EXO1, with several new inhibitors recently published. The confident visualization of the endogenous protein in IF, and certainly in IHC, has been historically difficult. If the authors are confident that this antibody is specifically recognizing EXO1 in cancer tissues based on their data, than showing the data would be beneficial to the larger community as a step towards proving this is a potential biomarker that can be assessed with existing reagents. However, the fact that it is a monoclonal, that should not be quantity limited, and has been apparently discontinued, makes me somewhat concerned as to why it is no longer available. If the authors are not confident enough to show primary data examples or controls using an antibody that is no longer available, then I think they should consider removing this data. While this data would provide evidence that EXO1 protein levels are indeed higher in actual cancer tissues, the paper as a whole does not hinge on this data that, as presented, raises numerous concerns.

Reviewer #2

(Remarks to the Author)

Response to referees

We would like to thank the reviewers for their helpful and constructive comments, which led to a significantly improved manuscript. To address the reviewers' concerns, we are submitting a substantially revised manuscript with 21 new figure panels.

We would also like to extend our gratitude for being allowed an extended revision timeline, as the first author of this study, graduate student Alexandra Nusawardhana, was out of the laboratory for an industry fellowship over the summer.

Reviewer #1

The manuscript examines the role of EXO1 overexpression in inducing genomic instability in BRCA positive breast cancers. The authors use the cBioPortal to interrogate TCGA datasets and find many cancers with amplifications and higher expression levels of EXO1, including around 20% of breast cancers. They then established HeLa and U2OS based EXO1 over-expressing cell lines to examine the phenotypic effects. They find that EXO1 overexpression leads to nascent strand degradation at ssDNA gaps and reversed forks using S1 nuclease fork assays and BrdU comet assays. They show using SIRF/PLA that active EXO1 localizes to nascent DNA in proximity to MRE11 and find that EXO1 OE leads to nascent DNA degradation at forks in HU treated cells that can be prevented by MRE11 inhibition or depletion of fork reversal proteins like SMARCAL1 and ZRANB3. Finally, they demonstrate using neutral comet and IF that EXO1 OE leads to DSBs and higher sensitivity to cisplatin, that they conclude may explain the differences in survival seen in TCGA data. Overall, they find that elevating EXO1 expression leads to nascent strand degradation in BRCA proficient cancers, in addition to its previously known activity in BRCA deficient cancers.

The group has a strong track record performing the types of assays used in the paper and the experimental data is for the most part well-presented and convincing. With this, they show that EXO1 overexpression via lentivirus causes a number of phenotypes similar to its previously established role in BRCA deficient settings in multiple cell lines. My primary issue with the paper is the clinical data analysis and its interpretation and whether the OE system is truly modeling what they imply. The overexpression system is based on the finding that EXO1 is frequently amplified/upregulated in breast cancer. This occurs primarily in the context of a large 1q43 amplification that contains a large number of other genes. Whether EXO1 protein levels are in fact elevated and if so, to an extent that is similar to what they assay experimentally in their lentiviral system, remains an open question. I therefore have some doubts about the the relevance of the experimental system and I have very strong reservations about their bioinformatic analysis and its interpretation. I have separated my comments into those related to bioinformatics and experiments.

We thank the reviewer for their helpful and encouraging comments.

We now show that EXO1 protein levels are increased in breast tumor tissues and cell lines compared to normal samples, and the overexpression obtained in our system is comparable to that found in breast cancer cell lines (new **Supplementary Figures S1B,C). This argues that our model does measure the impact of EXO1 expression in tumors. To provide further evidence that the mechanistic results obtained using this system are relevant, we now also show that breast cancer cell lines with higher endogenous EXO1 expression have increased nascent strand degradation compared to cell lines with lower endogenous EXO1 expression levels (new**

Supplementary Figures S1C, S3A, S3B) thus confirming the data obtained using the controlled overexpression system.

As detailed below, we use clinical data analysis to show that EXO1 expression correlates with genomic alterations in clinical samples. Beyond this, clinical data analysis, particularly in terms of patient outcomes, is not a critical aspect of our manuscript and we removed them. Instead, the **main point** of our manuscript is that EXO1 overexpression in tumors is associated with genomic instability, and the **mechanism** of this genomic instability is nascent strand degradation in BRCA-proficient cells. This is novel, as nascent strand degradation is not generally known to occur outside of deficiencies in the BRCA or TLS fork protection pathways. In the revised manuscript, we now show that EXO1 overexpression results in comparatively more MRE11 loading on reversed forks than BRCA2 knockdown, and BRCA2 depletion does not further increase MRE11 recruitment to these structures in EXO1-overexpressing cells (new **Figure 5H**). This indicates that EXO1 overexpression completely alleviates the fork protection activity of BRCA2, arguing that RAD51 loading on nascent DNA is not critical for fork protection, as previously thought, and instead suggesting that the BRCA proteins promote fork protection in a RAD51-independent manner. In our opinion, this is novel and important.

Major concerns

Bioinformatics

1. The authors stated: These findings argue that, despite the known role of EXO1 in DNA repair, increased EXO1 activity, rather than loss of EXO1 function, is associated with carcinogenesis. And later they state: Overall, these findings argue that increased EXO1 activity, rather than loss of EXO1 function, is associated with carcinogenesis, through increased genomic instability. This is a massive overinterpretation of the data. First, there is no indication in any sample that EXO1 activity, let alone its protein levels, are elevated. This could be addressed in tissue arrays by IHC or other methods but is not. Thus, this is an assumption based on mRNA or copy number. While it is possible that high EXO1 is more oncogenic than its loss, this data allows only a speculative point, as there is no evidence that amplified/overexpressed EXO1 is associated with clinical outcome and no evidence that it is responsible for the elevated genome instability. I make this point not to discount the experimental data in the paper, only that I think that this clinical data, that is very superficially analyzed, is being overinterpreted to make general statements that are unsupported by the data they are describing. If one wants to make prognostic claims about clinical data, at the very least, multivariate analyses on annotated clinical datasets should be used to calculate hazard ratios. BRCA deficient samples should also be separated from those datasets and the correlations with numerous other genes in the proposed pathway could also be assessed.

We agree with the reviewer that we have no indication that EXO1 activity is increased in patient tumors, and we removed the reference to activity from the manuscript. Studying the enzymatic activity of EXO1 in patient tumors is unfortunately not feasible. However, in the revised manuscript, we now investigate the EXO1 protein levels in breast cancer cell lines as well as in breast cancer tissue microarrays, as indicated by the reviewer. Overall, we find increased EXO1 protein expression in breast cancer samples compared to normal breast tissue (new **Supplementary Figures S1B,C**). Moreover, we show that the expression of the EXO1 protein in our overexpression system is similar to that found in some of these breast cancer cell lines (new **Supplementary Figures S1C**).

We agree with the reviewer that there is no direct evidence that amplified EXO1 is associated with clinical outcomes, and this is only speculation at this time. We have reframed our revised manuscript to clearly indicate this. We apologize if we were unclear and, potentially, unintentionally misleading when presenting our manuscript's significance. We are not claiming to show that increased EXO1 causes clinical outcomes. While this may be a possibility, as perhaps suggested by the survival data in the original Figure 7C,D, the currently available TCGA data are not strong enough to assess this. As pointed out by the reviewer in their comment #7, analyses of other databases or using other parameters can show the opposing trend. We thus removed this figure from the manuscript, along with any claim that our data shows an impact of EXO1 on patient outcomes.

Regarding the genomic instability aspect, on the other hand, we would like to respectfully argue that our data, overall, does indicate that EXO1 overexpression promotes genomic instability. There is a clear association between EXO1 levels and genomic instability in tumor datasets (**Supplementary Figures S2B-G**). This suggests that EXO1 overexpression may be responsible for this genomic instability. Since we cannot study it directly in patient tumors, we created the overexpression system in cell lines and show that EXO1 overexpression causes nascent strand degradation and subsequently double stranded DNA breaks. Mechanistically, this can account for the genome alterations observed in EXO1-overexpressing, BRCA-proficient tumors. This is the main point of our work, and we believe that the data presented support this model.

We agree that the statement that increased EXO1 activity, rather than loss of EXO1 function, is associated with carcinogenesis is a speculation. We are not drawing this conclusion from our results, but we believe that, generally speaking, our data support this possibility because we show increased genomic instability in EXO1-overexpressing cells, and increased genomic instability is known to cause carcinogenesis. Nevertheless, we agree that this is a speculation and have revised the manuscript accordingly (stating now that *These findings argue indicate that, despite the known role of EXO1 in DNA repair, increased EXO1 expression, rather than loss of EXO1 function, is found in cancer samples*).

2. Related to this point, a major confounding factor in the types of univariate analyses produced by cBioPortal, and many other online platforms, is proliferation. EXO1 is an E2F target and its expression highly correlated with a number of proliferation related genes (CHEK1, FOXM1, CCNA2 etc). Proliferation status often correlates with clinical outcomes, thus whether EXO1 per se is driving any clinical outcomes remains unclear, and is in fact argued against with the data used in Figure 7 (see next point on this). This, and other clinical parameters, need to be considered in survival data analysis using multiple variables

As mentioned above, we agree that it is difficult to demonstrate that EXO1 overexpression impacts clinical outcomes. While our data suggests that this may be the case, this is not the point of our manuscript. We have thus removed the patient outcomes data from our revised manuscript (Figures 7C,D in our original submission).

3. The authors stated: We found that breast cancer patients with altered EXO1 in tumors (which, as described above, represents amplification of EXO1 expression in the vast majority of cases) tend to have better survival than those with no EXO1 alterations (Fig. 7C,D). This is not what the data shows. First, this is a univariate analysis that is fine for hypothesis generation but not suited to interpret clinical data. Second, the p-values clearly indicate there is no significant

difference in survival. The differences that the authors are referring to are a handful of the samples (in one case 4 patients of >1000). This in no way justifies a statement like the one made.

As mentioned above, we agree that this point is difficult to make. This was a minor component of our original manuscript, and we now removed these analyses (Figures 7C,D in our original submission).

4. The authors stated: This is in line with the increased sensitivity of EXO1-overexpressing cells to cisplatin and suggests that EXO1 overexpression in tumors sensitizes them to cisplatin chemotherapy. Aside from the fact that these graphs show no difference in survival, how many of these patients received cisplatin?

The reviewer's comment illustrates the difficulty of using these datasets to draw meaningful conclusions. We do not know how many of these patients received cisplatin. This is why, we agree with the reviewer's comments on the lack of rigor of patient outcomes data and have removed it (Figures 7C,D in our original submission).

5. The EXO1 amplifications that are apparent in the dataset used are also seen in other well curated cohorts like METABRIC. These do not represent EXO1 amplification in isolation but a very large duplication of 1q43 that contains a number of other genes that may also influence oncogenesis. If you run the same analyses with any of these genes (for example MROH9, SDHC, CHRM3, RGS7) you get precisely the same outcome- that there is no statistical difference in survival and there is a correlation with genome alterations. While the experimental data would lead to a reasonable hypothesis that EXO1 amp may play a role in genome alterations, it does not account for all of the other alterations happening with EXO1 amp in any way.

We thank the reviewer for pointing this out. We agree with the reviewer that EXO1 expression seems to be part of a very large duplication of 1q43, which includes many other genes, and we are not investigating the other alterations.

Our aim is not to investigate the effect of this large scale amplification, but of EXO1 specifically. Even if the gene most relevant for carcinogenesis present on this amplification is not EXO1, and EXO1 amplification is simply a passenger event, this does not mean that its amplification does not have a functional impact on tumor cells. Many passenger events have been shown to contribute to cancer cell fitness, including in response to therapy (PMID: 28536279, PMID: 29875280, PMID: 32084333, PMID: 20876136). Thus, we do not believe that our data are any less relevant because EXO1 is amplified as a part of a large chromosomal arm event rather than as an individual amplification event. We are now discussing these issues in the revised manuscript.

6. The authors stated: EXO1 is overexpressed in a significant proportion of cancers, including in over 20% of breast tumors. First, the graph referred to in Figure 1A is based on a specific dataset and cannot be generalized to all breast cancer. One can look at this in cBioPortal across many different BC datasets and see variations, as each one has its own biases in tumor subtypes and other factors. Second, it is not actually an accurate description of the cohort

shown, as amplification is only shown in just over 5% of the Breast Cancer cohort (the actual cohort name should be used for proper reference), while the rest is mRNA high.

We made the changes in the text to reflect these nuances. We thank the reviewer for pointing them out.

7. The phrase “all breast cancers” is used repeatedly in Figure 1 while describing data related to a single cohort. The proportion of basal cancers with elevated EXO1 is higher in many of the breast cohorts, however it is worth noting there are a lot more of other subtypes that have EXO1 high expression or amplifications. In fact, the p-value for survival is lower in some of these cohorts with fewer basal cancers (ex. Metabric), calling into question the relevance to TNBC. However, I still maintain that this type of 1:1 correlation is not a serious way to establish what the authors wish to conclude. If you use another platform that conducts multivariate analysis on the same dataset, the answer is in fact the opposite (ex MammoOnc-DB), where you see statistically worse survival in the high EXO1 cohort. I would highly recommend the authors consult a statistician to do this analysis in a more rigorous way across multiple datasets to make conclusions better supported by the clinical data. I would also point out previous data (I am not vouching for its accuracy, only its existence) that proposes that 1q amps that contain EXO1 are associated with poor prognosis in breast cancer (<https://doi.org/10.1371/journal.pone.0077553>).

We thank the reviewer for this comment. We have updated the text referring to the original Figure 1 (Supplementary Figures S1 and S2 in the revised manuscript) to clearly indicate the subset and database used. Regarding the patient prognosis, as mentioned above, we agree that one cannot draw strong conclusions and we thus removed the patient outcomes data (Figures 7C,D in our original submission).

Experimental data

1. For Figure 2, it would be nice to see some representative data in the figure rather than just the quantification.

We now show representative figures for the BrdU alkaline comet assays (new **Figure 1D**).

2. All the experiments have been performed with EXO1 over-expression (WT/catalytically dead) in HeLa/U2OS cells. As EXO1 activity is shown to be dependent on MRE11, PRIMPOL, SMARCAL1 and ZRANB3, it would be reassuring to see that their relative levels are similar in the parental, WT and catalytically dead EXO1 lines.

We now show that the levels of these proteins are similar in parental, wildtype-overexpressing, and catalytic mutant overexpressing cells (new **Supplementary Figures S5A-D**).

3. The authors stated: This suggests that increased EXO1 levels results in recruitment of MRE11 to nascent DNA under replication stress through MRE11-EXO1 complex formation. Prior literature showed that the D173A mutant of EXO1 is catalytically dead, however it is capable of binding DNA as well as the WT, as shown previously by multiple groups (PMIDs:11842105, 40319035, 23178594) and it retains the PCNA binding PIP box (23939618). Has the localization of the WT and dead mutant been examined? It would seem possible that it is not simply levels, but levels and its activity that are needed.

We thank the reviewer for this suggestion. We investigated EXO1 recruitment to nascent DNA by SIRF experiments, under ssDNA gap-inducing conditions. We now show increased EXO1 recruitment to DNA in cells overexpressing wildtype EXO1, but not in those overexpressing the D173A catalytic mutant (new **Figures 4A-E**). While this mutant still binds DNA *in vitro* as pointed out by the reviewer, our results suggest that its retention on chromatin is impaired.

These experiments imply that, as indicated by the reviewer, increased EXO1 activity is driving MRE11-mediated nascent strand degradation by enhancing the localization of MRE11 to nascent DNA in BRCA proficient cells, and the co-localization between EXO1 and MRE11 at these structures. We speculate that EXO1 initiates degradation at the 5' end of the gap, expanding the gap and thus allowing more opportunities for MRE11 to engage the 3' end of the gap. This could drive gap expansion in BRCA-proficient cells, and subsequently genomic instability.

4. Additionally, the authors argue that “increased EXO1 levels result in recruitment of MRE11 to nascent DNA under replication stress through MRE11-EXO1 complex formation”. The MRN complex can bind DNA lesions in the absence of EXO1, so what more specifically are the authors proposing as the mechanism here? In the case of gaps, why is EXO1 needed for MRE11 to act in the other direction? While I would expect the resulting ssDNA to be shorter, I do not understand why MRN would not be able to act on these substrates without EXO1.

We agree with the reviewer that MRE11 can bind DNA lesions in the absence of EXO1, so it should act on these structures without EXO1. Indeed, we and others previously showed that MRE11 does act on ssDNA gaps in BRCA-deficient cells. However, in BRCA-proficient cells, the gaps are quickly filled by recombination-mediated gap filling. We now speculate that increased EXO1 activity may interfere with this process, by expanding the gap and thus providing more opportunities for MRE11 to engage the other end of the gap and bidirectionally expand it. This ultimately alleviates the ability of the BRCA pathway to fill the gap.

5. The authors stated: Thus, our study suggests that EXO1 overexpression causes a “BRCAness” phenotype, characterized by genomic instability and hypersensitivity to genotoxic chemotherapy. Are cells sensitive to PARP inhibitors?

We thank the reviewer for this suggestion. Indeed, in the revised manuscript, we now show that EXO1-overexpressing cells have increased sensitivity to PARP inhibitors (new **Figure 7B**).

Reviewer #2

We thank the reviewer for participating in reviewing the manuscript and for their useful comments.

Reviewer #3

The study investigates the role of the nuclease EXO1 in promoting genomic instability by degrading nascent DNA in cells that are proficient in the BRCA pathway. The researchers found that EXO1 is overexpressed in various cancers, leading to increased degradation of nascent DNA and double-strand break (DSB) formation, even in the presence of functional BRCA genes. EXO1 is overexpressed (EXO1 OE) in a significant proportion of cancers, including breast, hepatocellular, skin, testicular, and cervical cancers. This overexpression is associated with increased genomic instability.

EXO1 OE in BRCA-proficient cells leads to replication-associated DNA lesions upon replication stress. This is evidenced by increased single-strand DNA (ssDNA) gaps and reversed replication forks. EXO1 OE promotes the recruitment of MRE11 to nascent DNA, facilitating the degradation of ssDNA gaps and reversed forks. This interaction is critical for the observed genomic instability. The degradation of nascent DNA by EXO1 results in increased DSB formation. This leads to hypersensitivity to genotoxic agents like cisplatin. EXO1 OE sensitizes cells to chemotherapy, particularly in triple-negative breast cancer, which lacks hormone receptors and is typically treated with chemotherapy.

Overall, the findings highlight the dual role of DNA repair enzymes like EXO1 in maintaining genomic stability under normal conditions and promoting genomic instability when overexpressed. This manuscript is a good contender for Nature Communications.

We thank the reviewer for their encouraging and useful remarks.

Figure 2 : The alkaline comet assay also detects dsDNA breaks, wouldn't their assay then also detect replication-associated dsDNA breaks ? (like fork collapse) or do they consider these breaks negligible, also because of the low dose of HU ?

The reviewer is correct in pointing out that, technically, the alkaline comet assay can also detect dsDNA breaks. We have revised the text to reflect this possibility.

Figures 2C and 2D indicate that the EXO1 D173A mutant exhibits lower activity in U2OS cells compared to HeLa cells. This suggests that the role or recruitment/localization dynamics of EXO1 D173A may vary depending on the cellular context, highlighting potential cell line-specific differences in how the mutant functions. Has this been addressed experimentally ?

We have not consistently observed differences between the phenotypes of the EXO1 D173A mutant in HeLa compared to U2OS cells (see also our responses to the reviewer's comments below). In the revised manuscript, we investigated the localization of EXO1 in both HeLa and U2OS cells overexpressing wildtype or D173A EXO1 mutant. In both cell lines, the wildtype protein was found at nascent DNA gaps while the mutant was not (new **Figures 4A-E), with no apparent differences between the cell lines.**

The authors state that "MRE11 is recruited to nascent DNA in cells overexpressing EXO1 but not in cells overexpressing the catalytic mutant D173A, in both HeLa and U2OS cells." However, in Figure 4D, no significant difference in MRE11 recruitment is observed between EXO1 overexpression and EXO1 D173A overexpression in HeLa cells. Notably, there is also no

significant difference between the empty vector (EV) and EXO1 D173A conditions, raising concerns about the robustness of this conclusion in the HeLa context.

These differences between HeLa and U2OS cells are likely due to experimental variation. In the revised manuscript, we provided a new experiment showing that the D173A mutant abolishes MRE11 recruitment to nascent DNA to the same extent in HeLa and U2OS cells (new **Figure 3D**).

Figure 4D : The error bar in the EXO1 D173A condition is very high. Could the authors show individual N (like on Figures 7A & B) ? It appears that the EXO1 mutant is still able to recruit Mre11 to some extent. Could the authors comment on this?

As mentioned above, we now provide a new experiment showing that the mutant is unable to recruit MRE11 (new **Figure 3D**). Because of the large number of samples, it is impractical to show the individual values in the graph, but those values are listed in the Source Data file.

Figures 5C–G: How can the authors be certain that the SIRF signal they observe reflects degradation at inverted replication forks, especially without colocalization data using markers such as CtIP or MRE11?

The SIRF signal investigated in these figures is in fact the MRE11 signal. To demonstrate that this MRE11 SIRF signal indeed occurs at reversed replication forks reflecting their degradation, we now show that depletion of SMARCAL1, the translocase responsible for fork reversal, abolishes this MRE11 SIRF signal in EXO1-overexpressing cells (new **Figure 5H**).

Figure 5H: To strengthen the conclusion that degradation occurs at inverted forks via MRE11 activity, the authors should consider including an siMRE11 condition and assess for potential epistasis by performing a double siRNA knockdown.

In the revised manuscript, we show that depletion of MRE11 by siRNA, similar to its inhibition by mirin, fully suppresses the fork degradation observed in EXO1-overexpressing cells in both HeLa and U2OS cells (new **Supplementary Figure S4**). Since individual depletions of SMARCAL1, ZRANB3 or MRE11 all restore fork protection to the same extent as in control cells (CldU/IdU ratio of 1), co-depletion of any of them (by double siRNA knockdown) would result in the same phenotype (CldU/IdU ratio of 1). In other words, an additive phenotype would not be detectable, since the CldU/IdU ratio cannot get bigger than 1. Because of this, an epistatic relationship cannot be determined from such an experiment.

The authors should compare the levels of EXO1 Overexpression in a panel of cancer cell lines to those in their HeLa and U2OS cells: are they comparable?

We thank the reviewer for this comment. In the revised version, we now show that EXO1 expression is increased in breast cancer cell lines T47D, MDA-MB-231, MCF7 and MDA-MB-468 compared to MCF10A normal epithelial mammary gland cells. Moreover, we show that the EXO1 overexpression in our experimental system is comparable to that found in these breast cancer cell lines (new **Supplementary Figures S1C**).

The most impactful conclusion of the paper is that EXO1-overexpressing cells exhibit a BRCAness phenotype in terms of sensitivity to DNA damage agents; however, the authors don't thoroughly explore this aspect. Figures 7C and D only bring circumstantial evidence that EXO1-overexpressing cancers are more sensitive to cisplatin. I think Figure 7 (and the paper) would be greatly improved if the authors could examine cell survival following cisplatin treatment in a panel of cancer cell lines that overexpress or do not overexpress EXO1.

We thank the reviewer for this thoughtful comment. First of all, to enhance the relevance of our findings, we now show that EXO1-overexpressing cells are also sensitive to PARP1 inhibitors (new **Figure 7B**).

We appreciate the reviewer's suggestion to use the breast cancer cell lines to correlate cisplatin sensitivity with EXO1 expression levels. Unfortunately, because of the big genetic variation between the different cell lines grown *in vitro* for many years, affecting many different mechanisms of drug resistance, any EXO1-dependent effects may be masked. In fact, this is exactly why we decided to create our exogenous EXO1 overexpression system to begin with, instead of using breast cancer cell lines with different EXO1 expression.

This being said, we agree with the reviewer that it is important to show that EXO1 expression levels correlates with specific genome instability phenotypes in breast cancer cell lines. In the revised manuscript, using the DNA fiber combing assay we now show that T47D cells, which show the highest level of EXO1 expression in our hands, have increased replication fork degradation compared to MDA-MB-468 and MCF7 cells, which show comparatively lower EXO1 expression (new **Supplementary Figure S3**). (Unfortunately, because of varying replication rates and culturing conditions making them less amenable to the DNA fiber combing assay, we were not able to include the other cell lines in this analysis.) These findings support our model that EXO1 expression in tumors is associated with genomic instability through nascent strand degradation.

Additionally, a major point lacking in the discussion is that many publications have shown that EXO1 over-expression is associated with a poor prognosis and cancer progression, due to the increase in genome instability.

We thank the reviewer for this comment. As also explained in the answers to Reviewer #4, we acknowledge those studies. The issue of deriving patient prognosis conclusions from cancer datasets is a complicated one, as this is affected by many factors which are not known or well documented. Critically among these factors is whether the patients received treatment with genotoxic chemotherapeutic agents such as cisplatin. Not having that information to stratify the samples results in confounding findings. As explained in our answer to Reviewer #1, this is why we decided to remove the patient outcomes data (Figures 7C,D in our original submission).

The main point of our manuscript is that EXO1 overexpression in tumors is associated with genomic instability, and the mechanism of this genomic instability is nascent strand degradation in BRCA-proficient cells. This is novel, as nascent strand degradation is not generally known to occur outside of deficiencies in the BRCA or TLS fork protection pathways. We do not believe that any of the previous studies demonstrated that EXO1 overexpression causes genomic instability, and identified the mechanism of this genomic instability, as we do here.

Reviewer #4

EXO1 is an exonuclease with important roles in long-range resection at sites of replication stress or DNA double-strand breaks, generating single-stranded DNA that can then be engaged by homologous recombination repair or be subject to further nuclease attack, potentially leading to more damage.

The study starts off with showing that EXO1 mRNA levels are increased in cancer samples, such as from breast and cervical cancer. This finding is not new as it has been reported before by several groups (PMID: 24147022, PMID: 33623391, PMID: 39718579, PMID: 33569437). It is not investigated whether EXO1 protein levels are also increased in cancer, which has been attempted by others using breast cancer cell lines (PMID: 24147022).

The further premise of the study is that increased EXO1 expression in cancers could cause genome instability phenotypes that resemble those caused by loss of DNA repair factors such as BRCA1. The study goes on to show that ectopic EXO1 protein overexpression in model cancer cell lines causes phenotypes consistent with increased proximity of MRE11 nuclease to EXO1 and nascent DNA, increased resection at nascent DNA and increased DNA double strand break formation – if these cells are treated with reagents that induce replication stress. EXO1-overexpressing cells are potentially more sensitive to cisplatin, although the evidence for this is limited. The conclusion is that EXO1 overexpression might cause a BRCAness-like phenotype that could sensitise cancers to chemotherapy.

The experiments are of good quality and nicely controlled by using the catalytically inactive (nuclease-deficient) EXO1 D173A mutant. This shows that most phenotypes are caused by the EXO1 nuclease catalytic activity rather than overexpression per se. EXO1 D173A is still able to bind DNA and interact with other DNA repair factors, which helps explain why nuclease-independent effects are sometimes seen such as in Figure 3A or Figure 4D, F.

The mechanistic findings are generally convincing and useful for the field, but are limited in breadth, using a small number of assays that have not evolved much from the author's previous publications on EXO1 and MRE11. The experiments show that increased EXO1 activity causes certain DNA replication and DNA damage phenotypes that would be expected if high EXO1 activity is detrimental, but do not provide much new insight into EXO1 function. The study does not directly measure effects on genomic instability or whether EXO1 might be an oncogene. The cancer data analysis is too rudimentary to add anything to the already existing literature.

We thank the reviewer for their very useful comments.

Indeed, we are not directly measuring whether EXO1 may be an oncogene. EXO1-overexpressing animal models represent an important next step in studying the oncogenic effects of EXO1, but those costly, long-term studies are beyond the scope of the current manuscript. We sincerely hope the reviewer agrees with us.

Regarding genomic instability, we would like to respectfully argue that our manuscript does indicate that EXO1 overexpression promotes genomic instability: 1. We show that EXO1 overexpression causes nascent strand degradation and double strand break formation. (In the revised manuscript, we now provide additional evidence of this, by showing that breast cancer cell lines with increased EXO1 expression have increased nascent strand degradation compared to cell lines with lower EXO1 expression levels – new **Supplementary Figures S1C,**

S3A, S3B). 2. We show that EXO1 overexpressing cells are sensitive to cisplatin and PARP1 inhibitors (the PARPi sensitivity is shown in the revised manuscript in the new **Figure 7B**); **3.** We show that EXO1 overexpression in patient tumor samples correlates with increased rates of genome-wide alterations. While the increased expression of EXO1 in tumors was indeed shown before, to our knowledge the association with chromosome level genome alterations was not.

In addition, our work reveals critical novel insights into the **mechanism of fork protection**. In our opinion, nascent strand degradation is, in fact, not expected in BRCA pathway proficient cells, which are able to load RAD51 on reversed forks. For many years, this has been considered to represent the mechanism of fork protection, as RAD51 coating of reversed arms is supposed to block nuclease-mediated DNA resection. However, we find that EXO1 overexpression is enough to induce MRE11-mediated fork degradation, regardless of the BRCA status. In the revised manuscript, we now show that EXO1 overexpressing cells have more MRE11 loading to reversed forks than BRCA2-depleted cells (new **Figure 5H**). This shows that increased EXO1 activity promotes MRE11 engagement on reversed forks to a greater extent than loss of BRCA2-mediated RAD51 coating of reversed forks. Importantly, when we knocked down BRCA2 in EXO1-overexpressing cells, we found no additional increase in MRE11 loading. This epistatic interaction indicates that EXO1 overexpression completely alleviates the fork protection activity of BRCA2. These results argue that the presence of RAD51 on reversed forks is not enough to block MRE11-mediated fork degradation. Thus, our manuscript shows that RAD51 loading on nascent DNA is not critical for fork protection, as previously thought, and instead suggest that the BRCA proteins can promote fork protection in a RAD51-independent manner.

Specific comments:

Top of page 5: The results section starts with: “To investigate the relevance of EXO1-mediated genome instability in cancer, (...)”. This pre-supposes the findings of the study and does not explain why EXO1 expression levels in cancer were investigated to begin with. The authors could justify this much better. EXO1 is on located chromosome 1q, which is frequently amplified in breast cancer, and several aspects of high EXO1 expression in cancer have been characterised in previous studies (see above). This literature should be considered and cited.

We thank the reviewer for pointing out these studies, which we now cite, and we apologize for omitting to cite them in our original submission.

Bottom of page 6: “However, we found that EXO1 expression positively correlates with genome alterations (Fig. 1D), suggesting that EXO1 promotes chromosomal structural instability (...)” Correlation is not causation. High EXO1 expression could be a consequence of chromosomal instability.

We agree with the reviewer, which is why we used the term “correlates”. As explained above, our conclusion that EXO1 expression promotes genomic instability is based on the collective data we present, not only this correlation. For example, we show that EXO1 overexpression causes double stranded DNA breaks, a known intermediate through which chromosomal translocations occur.

Bottom of page 7: “Overall, these findings argue that increased EXO1 activity, rather than loss of EXO1 function, is associated with carcinogenesis, through increased genomic instability.” See

previous comment. It seems fair to say that increased EXO1 expression is associated with cancers with high genomic instability. It is not certain that EXO1 is highly expressed during cancer development or that it increases genomic instability.

We agree with the reviewer, and we changed this sentence to: "Overall, these findings argue that increased EXO1 levels may be associated with increased genomic instability in cancer samples."

Figure 4: The authors should elaborate more on why they think EXO1 would recruit MRE11 to nascent DNA.

*In the revised manuscript, we investigated EXO1 recruitment to nascent DNA by SIRF experiments, under ssDNA gap-inducing conditions. We now show increased EXO1 recruitment to DNA in cells overexpressing wildtype EXO1, but not in those overexpressing the D173A catalytic mutant (**Figure 4A-E**). These experiments imply that increased EXO1 activity is driving MRE11-mediated nascent strand degradation by enhancing the localization of MRE11 to nascent DNA in BRCA proficient cells, and the co-localization between EXO1 and MRE11 at these structures. We speculate that EXO1 initiates degradation at the 5' end of the gap, expanding the gap and thus allowing more opportunities for MRE11 to engage the 3' end of the gap. This could drive gap expansion in BRCA-proficient cells, and subsequently genomic instability.*

The labelling scheme in Figure 4A is not very convincing as a gap-specific assay. Data in Figure 3 support that gaps may be formed in the DNA that is synthesised during the treatment with 0.4 mM HU. But for the experiments in Figure 4A-E, nascent DNA is labelled in absence of HU and 0.4 mM HU is added afterwards, providing less opportunity for the MRE11 that co-localises with EdU to be at gaps. It would be helpful to show a SIRF experiment without any HU treatment to see the baseline in unchallenged cells and to support the conclusion that the effect in Figures 4D,E and 5F,G shows MRE11 recruitment specifically to gaps and arrested forks, respectively.

*We thank the reviewer for this useful comment. In the revised manuscript, we now show that in the absence of HU treatment, there is no recruitment of MRE11 to nascent DNA (new **Figure 3D**).*

Figure 7A, B: The link between the molecular findings in previous figures and cisplatin sensitivity would be more convincing if the sensitivity of EXO1 OE was normalised to the sensitivity of EXO1 D173A OE.

*We thank the reviewer for this useful comment. In the revised manuscript, we show that the overexpression of the catalytic mutant does not cause cisplatin sensitivity (new **Figure 7A**). Moreover, we also show that the overexpression of wildtype, but not of catalytic mutant EXO1 causes PARP1 inhibitor sensitivity (new **Figure 7B**).*

Materials and methods: Much more detail should be given on how the TCGA dataset analysis was performed and which datasets and search parameters used to generate Figure 1A and Figure 7C,D.

We now provide the requested information (since these analyses were moved to the Supplementary Information, we provide this information in the Legends to Supplementary Figures S1 and S2).

Response to referees

We would like to thank the reviewers for their helpful and constructive comments.

Reviewer #1

The authors have addressed many of the reviewer comments and included new data that helps support their original model and add additional depth. This included demonstrating that EXO1 activity is required for its retention, that its OE is epistatic with BRCA2 loss, that MRE11 is binding reversed forks with new SIRF data and that OE is roughly similar to some BC cell lines that are widely used. The overinterpretation of superficially analyzed clinical data has been toned down, and a subset has been moved to the supplement. My biggest reservation is that experiments are carried out for the most part in a single cell line background that is not a BC and has its own host of other genetic abnormalities, making some of the more generalized interpretations proposed difficult to support. While some data from cell lines expressing different levels of EXO1 is presented as further proof, this data is not particularly compelling.

We thank the reviewer for their comments. We would like to respectfully point out that our experiments are done in two different cell lines, from different backgrounds -namely U2OS osteosarcoma cells and HeLa cervical cancer cells. The large number of genetic variations present in cancer cell lines is exactly the reason why we chose to employ an isogenic system, by overexpressing EXO1 (or the catalytic mutant and the empty vector, as controls) in these cells. This system allows us to specifically investigate the impact of EXO1 overexpression, eliminating confounding effects caused by other genetic changes.

In addition, we provide correlative data using breast cancer cell lines with varying EXO1 expression, which supports our proposed model. In the revised manuscript, we provide additional replicate experiments using these breast cancer cell lines, as requested by the reviewer (see comment #1 below).

Specific comments-

1. Referring to this comment in the rebuttal- "by showing that breast cancer cell lines with increased EXO1 expression have increased nascent strand degradation compared to cell lines with lower EXO1 expression levels – new Supplementary Figures S1C, S3A, S3B)." The authors argue that they have demonstrated that nascent strand degradation is elevated in T47D that are high EXO1 expressors of the lines examined. The data in S3A and S3B does not appear to support this. In S3A there is a minor difference, but not in S3B, and no statistics employed anywhere here. MDA-MB-468 cells that potentially have more than two-fold OE of EXO1 and no effect on nascent strand degradation, making this data unsupportive of authors claims.

In the revised manuscript, we included two new, independent replicates of this nascent strand degradation experiment (new **Supplementary Figures S3C, S3D**). Moreover, we plotted the median values of the CldU/IdU ratios from each of the four experiments and performed statistical analyses. We show that nascent strand degradation is statistically significantly increased in T47D cells, which show the highest level of EXO1 expression, compared to MDA-MB-468 and MCF7 cells, which show comparatively lower EXO1 expression (new

Supplementary Figures S3E). We would like to point out that MDA-MB-468 and MCF7 cells do show nascent strand degradation (as shown by the fact the CldU/IdU ratio is less than 1), but not to the same extent as T47D cells.

2. The authors report a tissue microarray using EXO1 IF to determine levels. No primary data is provided to evaluate and no indication of what controls were carried out to ensure that the EXO1 antibody was on target are provided. The methods provided state that 2 different primary EXO1 antibodies were used. Please clarify these points.

We apologize but we do not understand why the reviewer states that “2 different primary EXO1 antibodies were used”. We clearly state that the EXO1 primary antibody Santa Cruz Biotechnology sc-56092 was used.

The experiment presented was performed on a commercially available tissue microarray slide (Novus NBP2-30212). We were not able to perform control experiments (eg. no primary antibody) since all samples were on the same slide, so they could not be differentially treated. We did perform control single antibody experiments but those were performed on a different tissue sample that was available to us. We observed minimal background signal in those control experiments. We are presenting below a representative micrograph showing this. We prefer to not include this in the manuscript, as the sample used for the test is different than the breast cancer microarray samples quantified in the figure. In addition, to this control, we would like to point out that the observed difference in EXO1 expression between breast cancer and normal tissue we observed, clearly indicates the specificity of the antibody.

Figure for Reviewer #1. Control tissue immunofluorescence experiment showing minimal background staining when the primary antibody was not included (blue - Hoechst staining; red - AF568 fluorescence).

3. In referring to Figure S2D as measuring genomic instability, the authors should clarify exactly what types of lesions are measured in this assessment, as it is relevant to the potential mechanism. Do these types of aberrations make sense with the proposed elevation in nascent strand resection based on other examples?

We thank the reviewer for this comment. The TCGA cBioPortal analysis defines Genome Alterations as large-scale copy number changes (deletions, amplifications, translocations).

These are events initiated upon double strand break formation. Since we show that nascent strand degradation correlates with double strand break formation (Figure 6), these findings align with our proposed model. We now discuss this in the revised manuscript.

4. In S2E-G, to me the most outstanding confounders are proliferation and the other genes amplified on 1q with EXO1. Do all 1q genes correlate with higher genome instability by this measure? Does proliferation? Looking quickly at CHK1 or KI67, it would appear so. The same applies to S2A, is this at all specific to EXO1 as is implied?

We have not analyzed all 1q genes for correlation with genome stability. As mentioned in the Discussion section, even if EXO1 was a “passenger” event, its overexpression may still have a functional impact. In other words, if expression of other genes on 1q correlate with genomic instability, this could still be caused by EXO1 overexpression. This is why, we do not think investigating all 1q genes for genomic instability in cancer samples would be that informative. Our experiments clearly indicate that EXO1 overexpression in cells causes genomic instability through nascent strand degradation, which is the main point of our manuscript.

(Regarding proliferation, as we discussed in the previous revision, we removed any data related to patient outcomes and are not considering proliferation in our analysis.)

5. In the first sentence of the discussion, and several other places, it is stated that EXO1 OE causes nascent strand degradation. However, in the absence of RS, there is no indication of elevated breaks or damage, so it would be worth clarifying these statements to some extent in that they are dependent on fork stalling, as nicely shown by a number of data points. I am not asserting that the authors are trying to misrepresent anything, I just think it is not always clearly stated. Particularly, considering that all the experiments are done biologically unrelated cell types and the BC cell lines seem to have no significant effect of EXO1 OE.

We thank the reviewer for this comment. We updated these statements to clearly indicate that the degradation is in response to replication stress (or at stalled/reversed/gapped forks).

6. I think it is a big leap to go from a HeLa cell model to proposing EXO1 as a biomarker generally for chemotoxic therapy. Is there any indication that it would outperform a proliferative marker for example? You would expect the same effects with a wide range of DNA damage types? This could be stated more specifically to align with the data.

We thank the reviewer for this comment. While this is ultimately just a speculation, we now clarify that EXO1 could be a biomarker for “replication stress-inducing cancer therapy”.

7. The authors stated: Since increased EXO1 expression is found in a higher proportion of cancers than BRCA inactivation, this has important implications for both cancer etiology as a prevalent mechanism of genomic instability, as well as cancer treatment, suggesting that EXO1 could be a biomarker for the response to genotoxic chemotherapy. Relevant to this, the authors do show new data indicating that there is PARPi sensitivity in the OE HeLa cells. However, this data is somewhat weak as a single dose is used and not compared to BRCA mutations (including epistasis) in this background.

In the revised manuscript, we added a new figure showing olaparib sensitivity with an additional dose (new **Figure 7C**). Moreover, we also included BRCA2-knockout cells in these new experiments. The olaparib sensitivity of EXO1-overexpressing cells was comparable to that of BRCA2-knockout cells. We were not able to perform epistasis experiments since it was not feasible to generate cell lines with concomitant EXO1 overexpression and BRCA2 knockout during this second round of manuscript revision, and we could not use BRCA2 downregulation by siRNA for this experiment, as cells are treated with olaparib for 10 days and siRNA-mediated downregulation is transient.

8. *The EXO1 SIRF experiments are performed with treatment of HU for just 20 mins (Fig 4 A-E) while the EXO1-MRE11 PLA has been done after 4hrs of HU. Did the authors have any rationale for selecting the two different time points, did authors test earlier time points for EXO1-MRE11 PLA? If so, mentioning that would be useful to further substantiate that EXO1 activity triggers MRE11 recruitment.*

The EXO1-MRE11 PLA experiments were performed using 3hr HU treatment, similar to the MRE11 SIRF, for which conditions were previously published (PMID: 29053959, PMID: 37805499). The EXO1 SIRF was performed using conditions our laboratory recently published (PMID: 38180818). Because of the 5'-3' directionality of EXO1 (as opposed to the 3'-5' directionality of MRE11), we think that longer HU treatment conditions may result in removal of EXO1 from DNA as the 3' end is reached. We agree with the reviewer that additional time points for the EXO1-MRE11 PLA experiments would be useful, but these experiments are complicated and time consuming, and in any case the HU treatment conditions may not translate similarly between the PLA and SIRF assay (which also require EdU incorporation).

Reviewer #2

We thank the reviewer for participating in reviewing the manuscript and for their useful comments.

Reviewer #3

The authors have satisfactorily addressed all my comments, and the manuscript is now acceptable for publication.

We thank the reviewer for their comments and for their helpful suggestions during the manuscript revision.

Reviewer #4

The authors have addressed my comments to my satisfaction.

We thank the reviewer for their comments and for their helpful suggestions during the manuscript revision.

Response to referees

We would like to thank the reviewers for their helpful and constructive comments.

Reviewer #1

The authors have addressed the majority of my concerns and the expanded data in Figure 7 and Figure S3 add more confidence to the conclusions. I remain concerned about the data in S1B due and have to reiterate one of my primary concerns that was ignored in the response- no primary data is provided to evaluate.

2. The authors report a tissue microarray using EXO1 IF to determine levels. No primary data is provided to evaluate and no indication of what controls were carried out to ensure that the EXO1 antibody was on target are provided. The methods provided state that 2 different primary EXO1 antibodies were used. Please clarify these points.

We apologize but we do not understand why the reviewer states that “2 different primary EXO1 antibodies were used”. We clearly state that the EXO1 primary antibody Santa Cruz Biotechnology sc-56092 was used.

I think due to the wording, I thought the Novus product was also an antibody, thanks for clarifying that.

The experiment presented was performed on a commercially available tissue microarray slide (Novus NBP2-30212). We were not able to perform control experiments (eg. no primary antibody) since all samples were on the same slide, so they could not be differentially treated. We did perform control single antibody experiments but those were performed on a different tissue sample that was available to us. We observed minimal background signal in those control experiments. We are presenting below a representative micrograph showing this. We prefer to not include this in the manuscript, as the sample used for the test is different than the breast cancer microarray samples quantified in the figure. In addition, to this control, we would like to point out that the observed difference in EXO1 expression between breast cancer and normal tissue we observed, clearly indicates the specificity of the antibody.

The SC antibody listed seems to be discontinued, but archived data sheets do not mention IHC as a tested application or show examples of IF or tissue staining. If a tissue microarray was stained and scored, that “clearly indicates the specificity of the antibody”, I think providing some example images of what was actually scored is a very reasonable request that was ignored in the author response. I also think that the case made for the control data is reasonable and do not understand why it would not be shown, absent any other more rigorous controls (we typically prepare FFPE blocks using KO cell line pellets for example).

There is currently a lot of clinical interest in EXO1, with several new inhibitors recently published. The confident visualization of the endogenous protein in IF, and certainly in IHC, has been historically difficult. If the authors are confident that this antibody is specifically recognizing EXO1 in cancer tissues based on their data, than showing the data would be beneficial to the larger community as a step towards proving this is a potential biomarker that can be assessed with existing reagents. However, the fact that it is a monoclonal, that should not be quantity

limited, and has been apparently discontinued, makes me somewhat concerned as to why it is no longer available. If the authors are not confident enough to show primary data examples or controls using an antibody that is no longer available, then I think they should consider removing this data. While this data would provide evidence that EXO1 protein levels are indeed higher in actual cancer tissues, the paper as a whole does not hinge on this data that, as presented, raises numerous concerns.

We thank the reviewer for the supportive comments. We agree that the paper as a whole does not hinge on the data presented in Supplementary Figure S1B. This data (breast tissue staining) agrees with the bioinformatic analysis (Supplementary Figure S1A) and the cell lines western blots (Supplementary Figure S1C). Nevertheless, considering the reviewer's comments, we are now removing Supplementary Figure S1B from the manuscript, as suggested by the reviewer.

Reviewer #2

We thank the reviewer for participating in reviewing the manuscript and for their useful comments.